# Associations between physical activity and CVD-related metabolomic and proteomic biomarkers

**Örjan Ekblom**[1,2*], **Harry Björkbacka**[3], **Mats Börjesson**[4,5], **Elin Ekblom-Bak**[1],
**Anders Blomberg**[6], **Kenneth Caidahl**[7,8], **Ewa Ehrenborg**[9], **Gunnar Engström**[3],
**Jan Engvall**[10,11], **David Erlinge**[12], **Tove Fall**[13], **Bruna Gigante**[14,15], **Anders Gummesson**[16,17],
**Tomas Jernberg**[18], **Lars Lind**[19], **David Molnar**[17,20], **Jonas Oldgren**[21,22], **Aidin Rawshani**[17,23],
**Johan Sundström**[24,25], **Stefan Söderberg**[6], **Patrik Wennberg**[26], **Carl Johan Östgren**[10,27]

1 Department of Physical Activity and Health, The Swedish School of Sport and Health Sciences (GIH), Stockholm, Sweden, 2 Department of Neurobiology, Care Sciences and Society, Division of Nursing, Karolinska Institutet, Stockholm, Sweden, 3 Department of Clinical Sciences in Malmö, Lund University, Malmö, Sweden, 4 Center for Lifestyle Intervention, Department of Molecular and Clinical Medicine, Institute of Medicine, Sahlgrenska Academy, University of Gothenburg, Gothenburg, Sweden, 5 Sahlgrenska University Hospital, Region Västra Götaland, Gothenburg, Sweden, 6 Department of Public Health and Clinical Medicine, Umeå University, Umeå, Sweden, 7 Department of Clinical Physiology, Karolinska University Hospital, and Karolinska Institutet, Stockholm, Sweden, 8 Department of Clinical Physiology, Sahlgrenska University Hospital, and Sahlgrenska Academy, Gothenburg, Sweden, 9 Cardiovascular Medicine Unit, Department of Medicine, Center for Molecular Medicine, Karolinska Institute and Karolinska University Hospital, Stockholm, Sweden, 10 CMIV, Centre of Medical Image Science and Visualization, Linköping University, Linköping, Sweden, 11 Department of Clinical Physiology, and Department of Health, Medicine and Caring Sciences, Linköping University, Linköping, Sweden, 12 Department of Clinical Sciences Lund, Cardiology, Lund University, and Skåne University Hospital, Lund, Sweden, 13 Department of Medical Sciences, Molecular Epidemiology, Uppsala University, Uppsala, Sweden, 14 Division of Cardiovascular Medicine, Department of Medicine, Karolinska Institutet, Stockholm, Sweden, 15 Department of Cardiology, Danderyds Hospital, Stockholm, Sweden, 16 Department of Clinical Genetics and Genomics, Region Västra Götaland, Sahlgrenska University Hospital, Gothenburg, Sweden, 17 Department of Molecular and Clinical Medicine, Institute of Medicine, Sahlgrenska Academy, University of Gothenburg, Gothenburg, Sweden, 18 Department of Clinical Sciences, Danderyd University Hospital, Karolinska Institutet, Stockholm, Sweden, 19 Department of Medical Sciences, Clinical Epidemiology, University of Uppsala, Uppsala, Sweden, 20 Department of Radiology, Region Västra Götaland, Sahlgrenska University Hospital, Gothenburg, Sweden, 21 Department of Medical Sciences, Cardiology, Uppsala University, Uppsala, Sweden, 22 Uppsala Clinical Research Center, Uppsala University, Uppsala, Sweden, 23 Wallenberg Laboratory for Cardiovascular and Metabolic Research, Institute of Medicine, University of Gothenburg, Gothenburg, Sweden, 24 Department of Medical Sciences, Uppsala University, Uppsala, Sweden, 25 The George Institute for Global Health, University of New South Wales, Sydney, Australia, 26 Department of Public Health and Clinical Medicine, Family Medicine, Umeå University, Umeå, Sweden, 27 Department of Health, Medicine and Caring Sciences, Linköping University, Linköping, Sweden

* orjan.ekblom@gih.se



## Abstract

### Aim

Habitual physical activity (PA) affects metabolism and homeostasis in various tissues and organs. However, detailed knowledge of associations between PA and cardiovascular disease (CVD) risk markers is limited. We sought to identify associations between accelerometer-assessed PA classes and 183 proteomic and 154 metabolomic CVD-related biomarkers.

**Data availability statement:** The General Data Protection Regulation (EU 2016/679) classifies de-identified versions of sensitive data that are sufficiently detailed to allow for re-identification as sensitive personal information. According to Swedish law (Law 2003:460 for ethical review of research involving humans), ethical permission is required to process such data. In accordance with Swedish legislation, the data can and will be made available to researchers who meet the criteria for access to confidential data, which includes obtaining their own ethics approval from the Swedish Ethical Review Authority (email: registrator@etikprovning.se; website: https://etikprovningsmyndigheten.se). Data applications can then be made by contacting SCAPIS (email: scapis@scapis.org; website: https://www.scapis.org/data-access/).

**Funding:** ÖE was funded by Livförsäkringsbolaget Skandia, Risk&Hälsa in the form of salary. The funders had no role in study design, data collection and analysis, decision to publish, or preparation of the manuscript.

**Competing interests:** J. S. reports direct or indirect stock ownership in companies (Anagram kommunikation AB, Sence Research AB, Symptoms Europe AB, MinForskning AB) providing services to companies and authorities in the health sector including Amgen, AstraZeneca, Bayer, Boehringer, Eli Lilly, Gilead, GSK, University of Gothenburg, Itrim, Ipsen, Janssen, Karolinska Institutet, LIF, Linköping University, Novo Nordisk, Parexel, Pfizer, Region Stockholm, Region Uppsala, Sanofi, STRAMA, Takeda, TLV, Uppsala University, Vifor Pharma, WeMind. S.S. has received compensation for consultancy or speaking for Johnson & Johnson and Merck. The rest of the authors declared that no competing interests exist.

## Method

We utilized cross-sectional data from the main SCAPIS cohort (n = 4647, median age: 57.5 yrs, 50.5% female) as a discovery sample and the SCAPIS pilot cohort (n = 910, median age: 57.5 yrs, 50.3% female) as a validation sample. PA was assessed via hip-worn accelerometers, while plasma concentrations of proteomic biomarkers were measured using Olink CVD II and III panels. Metabolomic markers were assessed using the Nightingale NMR platform. We evaluated associations between four PA classes (moderate-to-vigorous PA [MVPA], low-intensity PA [LIPA], sedentary [SED], and prolonged SED [prolSED]) and biomarkers, controlling for potential confounders and applying a false discovery rate of 5% using multiple linear regressions.

## Results

A total of eighty-five metabolomic markers and forty-three proteomic markers were validated and found to be significantly associated with one or more PA classes. LIPA and SED markers demonstrated significant mirroring or opposing relations to biomarkers, while prolSED mainly shared relations with SED. Notably, HDL species were predominantly negatively associated with SED, whereas LDL species were positively associated with SED and negatively associated with MVPA. Among the proteomic markers, eighteen were uniquely associated with MVPA (among those Interleukin – 6 [IL6] and Growth/differentiation factor 15 [GDF15] both negatively related), seven with SED (among those Metalloproteinase inhibitor 4 [TIMP4] and Tumor necrosis factor receptor 2 [TNFR2], both positively related), and eight were related to both SED/prolSED (among those Lipoprotein lipase [LPL] negatively related to SED and leptin [LEP] positively related to SED) and MVPA (with LPL positively related to MVPA and LEP negatively related to MVPA).

## Conclusion

Our findings suggest the existence of specific associations between PA classes and metabolomic and cardiovascular protein biomarkers in a middle-aged population. Beyond validation of previous results, we identified new associations. This multitude of connections between PA and CVD-related markers may help elucidate the previously observed relationship between PA and CVD. The identified cross-sectional associations could inform the design of future experimental studies, serving as important outcome measures.

## Introduction

Analyses of single plasma or serum biomarkers have increased our understanding of how PA may relate to, for example, CVD risk [1]. Still, analysing single or a few biomarkers is probably not enough to capture the full width of the processes associated with PA [2]. Simultaneous assessments of many hundred CVD-related

biomarkers have been made possible using proteomics and metabolomics. Only a limited number of previous studies have been published using these techniques, but results can potentially shed light on the variation between individual biomarkers in cardiometabolic responses to PA. Stattin et al. [3] reported a relation between self-reported PA and 28 cardiovascular-specific proteins related to several atherosclerotic processes such as low-density lipoprotein oxidation, protein degradation, and immune cell adhesion and migration. However, self-reported PA is often less temporally granulated compared to accelerometer-based data, which hinders comparisons between time spent in various intensities of PA (i.e., moderate or intense PA vs sedentary behaviours or low-intensity PA). Further, a low-precision PA measurement with larger random and systematic errors, such as self-reports compared to sensor-based measures used as an exposure variable [4,5], will generate regression dilution [6] and thereby under-estimate associations. Accelerometer-based PA metrics have been shown to possess a stronger relation to CVD-related biomarkers, compared to that of self-reports [7]. Also, accelerometer-based variables have been shown to be highly predictive for all-cause and CVD mortality [8,9].

Using accelerometer-based PA data and NMR lipidomic, Henson et al. reported unfavourable relations between time spent sedentary and several markers (including VLDL, HDL, and apolipoprotein A1) in 509 middle-aged participants at high risk of type II diabetes. In younger men, Vaara et al. [10] reported only a few associations between accelerometer-based PA and IDL triglycerides 3triglycerides 3-hydroxybutyrate, indicating that associations are weaker in younger individuals. To what extent these associations are present in a mixed, middle-aged population is unknown.

Time spent sedentary (SED) and in moderate to vigorous physical activity (MVPA) is typically not strongly related, but SED and low-intensity physical activity (LIPA) are often strongly, negatively correlated [10]. As there is an interrelation between the different PA classes, such that for example time spent in MVPA may alter the relation between time spent in SED and CVD risk [11], it is important to be able to adjust analyses accordingly. There are several studies on proteomic and metabolomic biomarkers in relation to PA, based on either clinical populations [12], effects from short-term high-intensity interventions [13–15],or based on self-reported physical activity or fitness [16]. These studies generally report beneficial relations to PA and limited, but detrimental, relations to time spent sedentary. Studies on PA in relation to proteomics are less well understood, as many proteins are involved in multiple pathways, but data indicate a beneficial relation between PA and inflammatory markers and markers for lipid metabolism. However, there is a paucity of studies reporting relations between both proteomic and metabolomic biomarkers and sensor-based PA, in the same cohorts and applying a cross-validation design [17].

Regarding sedentary time, there is evidence to suggest that longer periods of uninterrupted sitting have a separate association with CVD health compared to short bouts of sitting or other physically inactive events [18]. Therefore, there may be unknown differences in associations between CVD-related biomarkers and different PA classes. This lack of data on which biomarkers or groups of biomarkers are related to PA classes limits the understanding of potential genetic or behavioural contributions. As previous studies have indicated causal relationships between PA to sarcopenia [19] and metabolic disorders [20], there is a need for such data in designing individualized preventative efforts including PA.

The aim of the present study was therefore to explore and validate associations between proteomic (Olink) and nuclear magnetic resonance metabolomic (NMR) CVD-related biomarkers and four different accelerometer-based PA classes in two population-based samples of middle-aged individuals in Sweden. We hypothesize that relations between PA and biomarkers are different across PA classes.

## Materials and methods

This study was based on cross-sectional data from the Swedish CArdioPulmonary bioImage Study (SCAPIS) and included data from the main cohort as well as the local Gothenburg pilot study. We used data from the main SCAPIS cohort (n = 5557) as a discovery sample and data from the SCAPIS pilot cohort (n = 910) as a validation sample. Bergström et al have described the SCAPIS cohorts and data collection in detail [21].

### SCAPIS main cohort (discovery sample)

The SCAPIS main cohort is a Swedish cohort of randomly selected participants from the general population (age 50–64 yrs) in six regions in Sweden. The SCAPIS data collection was conducted between February 14th, 2013, and October 30th, 2018, as a multicentre study at Swedish university hospitals.

### SCAPIS pilot cohort (validation sample)

The Swedish CArdioPulmonary bioImage Study (SCAPIS) pilot study, which was conducted at the Sahlgrenska University Hospital in Gothenburg, Sweden, between March 12th, 2012, and October 26th, 2012, was used as a validation sample.

The data collection for the pilot study and the main study were approved by the ethics boards at Gothenburg University (No 638−16) and a Umeå University (No 2010–228-31M and 2017–183-31M), respectively. All data collection adheres to the Declaration of Helsinki and all participants gave written consent.

### Assessment of the PA classes

Accelerometer-based sedentary behaviours and PA patterns were derived from tri-axial accelerometers (ActiGraph model GT3X and GT3X+, ActiGraph LCCo, Pensacola, FL, USA). The participants wore the accelerometers for 7 days. Participants were instructed to wear the accelerometer in an elastic belt over the right hip during all waking hours, except during water-based activities, and to return after the wearing period. ActiLife v.6.13.3 software was used to initialize the accelerometers and to download and process the collected data. The accelerometer recorded raw data (sample rate set to 30 Hz) from three axes, which were combined into a resulting vector, expressed in counts per minute (cpm). In the main SCAPIS cohort data were down-sampled and extracted as 60-second epochs using a low-frequency extension filter. The use of a low-frequency extension filter may affect the time spent in lower intensities, but the relations to biomarkers are influenced to a minimal degree. For the pilot SCAPIS sample, the data available were extracted as 15-second epochs.

In both samples, non-wear time was defined as sixty or more consecutive minutes with no movement (0 cpm), with allowance for a maximum of 2 minutes of counts between 0 and 199 cpm. Wear time was calculated as 24 hours minus non-wear time. Participants with a minimum of 600 minutes of valid daily wear time for at least 4 days were included [22].

Wear time was calculated as 24 hours minus non-wear time. Sedentary time was defined as <200 cpm [23], low-intensity PA (LIPA) as 200–2689 cpm, and moderate-to-vigorous intensity PA (MVPA) as ≥2690 [24]. A prolonged sedentary (prolSED) bout was defined as ≥20 min of cpm below 200 [25], with no allowance for interruption above the threshold. As daily wear time varied between study participants above the minimum of 600 min per day, the PA pattern is presented as a percentage of wear time spent in different intensity-specific categories.

### Blood sampling and analysis

In the discovery sample, all sites contributed 850 samples each using random numbers, except for Stockholm, which contributed 825 samples, giving a total of 5075 samples. Discovery data initially included 30,154 subjects, and after the exclusion of subjects with completely missing biomarker data (n=25079) and completely missing PA variables (n=188), a total of 4647 subjects remained. Validation data initially included 1111 subjects, and after the exclusion of subjects with completely missing biomarker data (n=43) and completely missing PA variables (n=147), a total of n=921 remained. In total 240 subjects in the discovery data had more than 5% missing proteins, and 11 subjects in validation data had more than 5% missing proteins. These subjects were removed from the data, leaving 4647 and 910 subjects in the two samples, respectively.

Samples were included consecutively starting from the earliest date possible for each site to optimize the potential of prospective studies. Blood samples were bio-banked before analyses.

Samples subsequently underwent proteomic profiling using the Olink Proseek Multiplex Cardiovascular II and III (CVD II and CVD III) panels (Olink Proteomics, Uppsala, Sweden), analysing a total of 183 protein biomarkers. Samples further underwent metabolomic profiling using a high-throughput NMR metabolomics platform (Nightingale) analysing 154 markers. Markers identified in proteomic and metabolomic profiling are reported separately, due to their differing methodology and units of expression (mmol/L for NMR and relative normalized protein expression [NPX] quantification for Olink) that limit comparisons.

Names of the metabolites and proteins are given in Tables 2–4.

## Confounders

Sex was recorded binary as male and female using self-report. Age at assessment was recorded in years with one decimal. Kidney function was assessed as estimated glomerular filtration rate (eGFR) was calculated according to the Cockcroft-Gault equation and based on plasma creatinine. Diabetes was classified as No diabetes (fasting p-glucose <6,1 mmol/l and HbA1c < 42 mmol/mol), IFG (fasting p-glucose ≥6,1 and < 7,0 mmol/l and/or HbA1c > 42 and < 48 mmol/mol), or Diabetes (Diabetes stated by study participant at medical history interview or in subject questionnaire, or fasting p-glucose =>7,0 mmol/l or HbA1c =>48 mmol/mol).

Self-reported data on education (university degree vs no university degree) and smoking exposure (as pack-years) were collected. Diet was not standardized or controlled, but blood samples were drawn in a fasted state and intakes of carbohydrates, fats, and proteins (in grams per day, with calculations made from the MiniMeal-Q questionnaire) were used as potential confounders, together with measured body mass (in kilograms), and waist circumference (in centimetres).

## Statistics

Subject characteristics are reported as median (IQR) and proportions (IQR) for continuous and categorical variables, respectively.

The aim was to explore any associations between biomarkers and each of the four PA classes. The analyses were divided into three steps.

In the first step, unadjusted (crude) analyses were performed, with the biomarker as the dependent variable and each PA class as independent. Biomarker concentrations were transformed to the log2 scale.

In the second step, the potential confounders sex, age, eGFR, and Diabetes were added to the model. For proteomics, CVD II, and CVD III biomarkers plate number was also included as a confounder. Normal linear regression models were estimated and each p-value for the PA variables was stored. The p-values are then sorted and following the FDR procedure (Benjamini-Hochberg correction) converted into q-values. Q-values can in principle be interpreted similarly to p-values. No imputation was performed for missing data or outliers. This procedure was performed on the discovery dataset. A cut-off of 5% for the q-values was used and for the biomarkers where the q-values fell below 5%, the model was re-estimated on the validation dataset. For the validation dataset, the p-values were used to establish significance (5% level used)

The biomarkers that were significant for the validation set in the second step were included in the third step.

In the third step, the same model as in Step 2 was used, but more potential confounders were included (education, smoking status, body mass, waist circumference, intake of carbohydrates, fats, and proteins). Also, for SED, prolSED, and LIPA, the variable MVPA was included, and for MVPA, the variable SED was included. The model for step 2 was estimated for both discovery and validation datasets and used p-values to establish significance. Beta coefficients from the linear regressions are presented when associations were significant at step 1 for both validation and discovery datasets and in model 2 in the discovery dataset.

The included variables all have their theoretical significance, but unadjusted beta coefficients along with those from the two regression models were applied for transparency.

We explored interactions regarding sex and waist circumference (dichotomized at 88 cm and 102 cm for females and males, respectively). Directions of associations are given as (+) for positive and (-) for negative associations, respectively. Venn diagrams were generated according to Hulsen et al [26].

To separate biomarkers with a more solid ("robust") and more discrete ("separate") relation to PA beyond the application of two-sample validation and two different regression models, "robust relations" were considered for markers inversely and significantly related to both MVPA and SED/prolSED. "Separate relations" were considered when a marker was significantly related to MVPA or SED/prolSED and with some marginal not related to another (p-value>0.1). In Figs 1 and 2, biomarkers with separate but not robust relations are in parentheses. Potential non-linearity for the association between significant biomarkers and PA classes MVPA and SED was explored using a second-degree term to model 2.

## Results

Participant characteristics are shown in Table 1. Participants in the discovery and validation samples differed in some respects. Time spent in SED and in prolSED were higher, and LIPA lower, in the validation sample. A lower proportion of participants in the validation sample had a university degree, while sex distribution and age did not differ.

Of all included markers, 193 were significantly related to SED in model 1 in the validation sample. Corresponding numbers were 176, 155, and 215 for prolSED, LIPA, and MVPA, respectively. A total of eighty-five metabolomic and forty-three proteomic biomarkers were significantly related to one or more PA classes in model 1 in both validation and discovery samples and in model 2 for the discovery sample (Figs 1 and 2). LIPA and SED almost completely shared related

**Table 1. Participant characteristics in the two samples.**

| Variable | Total (N = 5557) | Discovery (N = 4647) | Validation (N = 910) |
|---|---|---|---|
| | *Median (IQR) or n (%)* | *Median (IQR) or n (%)* | *Median (IQR) or n (%)* |
| SED (% of wear time) | 56 (48-64) | 54 (46-61) | 69.7 (64.6-74.4) |
| prolSED (% of wear time) | 186 (118-277) | 164 (107.5-232) | 362.1 (283.2-453.6) |
| LIPA (% of wear time) | 38 (30-45) | 40 (33-46) | 25.1 (20.9-29.4) |
| MVPA (% of wear time) | 6 (4-8) | 6 (4-8) | 4.8 (3.3-6.6) |
| Daily wear time (min) | 875 (824-924.6) | 876 (825-925) | 869 (819-921.9) |
| Sex | | | |
| Female, n (%) | 2807 (50.5%) | 2349 (50.5%) | 458 (50.3%) |
| Male, n (%) | 2750 (49.5%) | 2298 (49.5%) | 452 (49.7%) |
| Age (yrs) | 57.5 (53.9-61.3) | 57.5 (53.9-61.3) | 57.5 (53.8-61.7) |
| Education | | | |
| University degree, n (%) | 2478 (44.8%) | 2132 (46.0%) | 346 (38.2%) |
| eGFR | 118.1 (100.9-138.3) | 116.9 (100-137) | 123.8 (105.3-144.8) |
| Diabetes | | | |
| Normoglycemia, n (%) | 4371 (78.7%) | 3654 (78.6%) | 717 (79.2%) |
| IFG, n (%) | 813 (14.6%) | 685 (14.7%) | 128 (14.1%) |
| Diabetes, n (%) | 367 (6.6%) | 307 (6.6%) | 60 (6.6%) |
| Smoking | | | |
| Pack-year | 12.8 (5.3-23) | 12.3 (5.3-22.5) | 15 (6-25.2) |
| Body mass (kg) | 65 (56-74) | 65 (56-74) | 64 (55-73.2) |
| Waist circumference (cm) | 94 (85-102) | 93 (85-102) | 95 (86-102) |
| CHO intake (g/d) | 173.6 (128.7-229.4) | 170.1 (126.2-223) | 194.6 (140.1-263.7) |
| Fat intake (g/d) | 64 (47.9-84.3) | 63.7 (47.8-83.7) | 65.2 (49.5-88) |
| Protein intake (g/d) | 65.4 (50.9-82.9) | 64.6 (50.4-81.9) | 70.2 (54.3-88.9) |

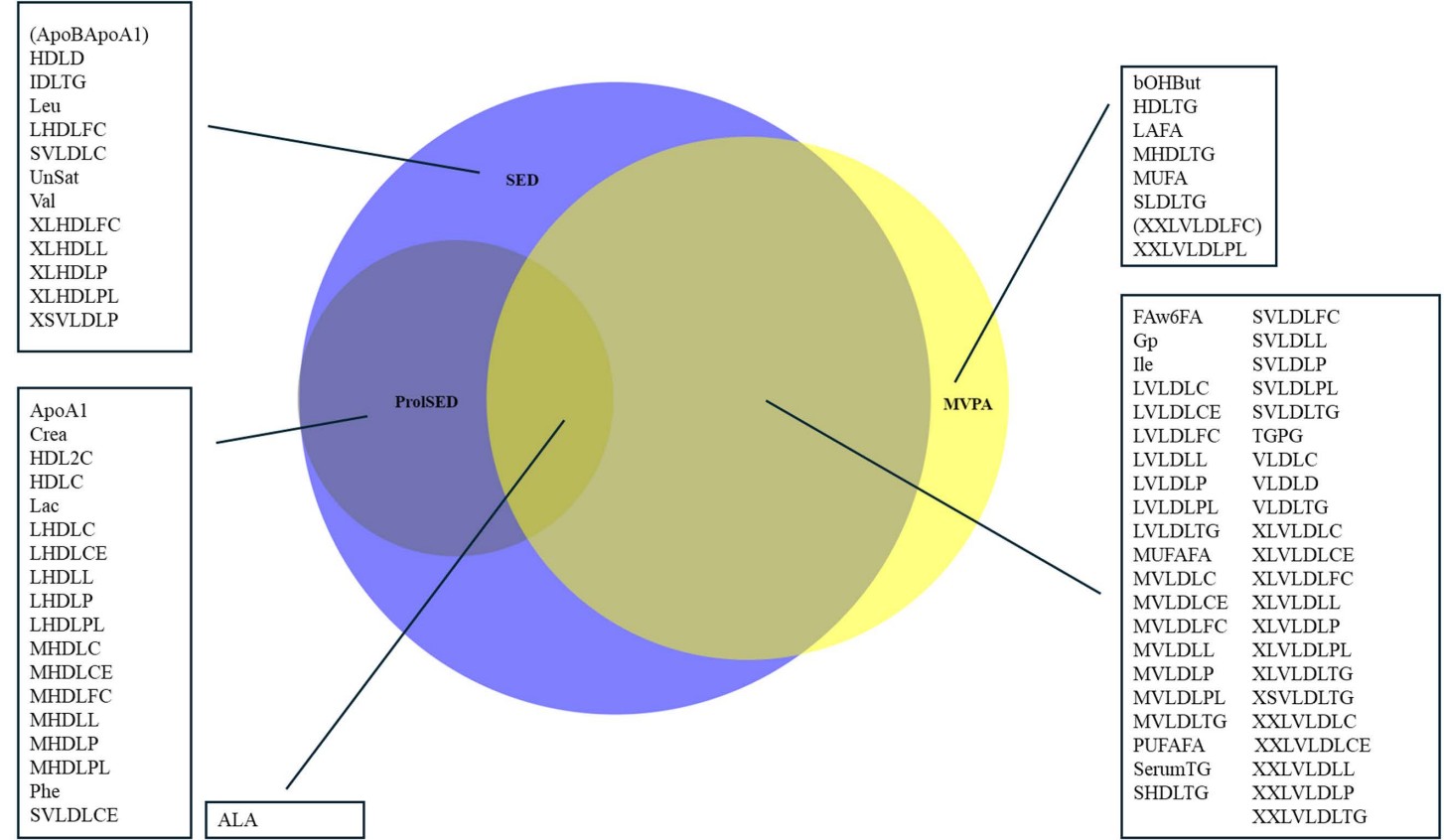

**Fig 1. Proportional Wenn-diagram for metabolomic markers significantly related to the three PA classes, SED (light blue), prolSED (dark blue), and MVPA (yellow).**

metabolomic and proteomic markers. LIPA shared all of its related markers with SED, and beta coefficients were close to opposite each other, which may be explained by the strong negative interrelation (rho = −0.94) between SED and LIPA. (S1 and S2 Figs) Arbitrarily, LIPA was excluded from the analyses for clarity.

## Metabolomic markers

Of the eighty-three metabolic markers, eight were related to solely to MVPA, and thirteen to SED, with the remaining being related to two or three PA classes (Fig 2). MVPA and SED, the two most differing PA classes intensity-wise, shared forty-three metabolomic markers, mainly VLDLs. ProlSED and SED shared 19 metabolomic markers, primarily HDLs. Of all eighty-three markers, only eight were not related to SED. Beta coefficients were nominally low (S3 Fig, Tables 2–4]. We found the nominally strongest associations between MVPA and LAFA (+), FAw6FA (+), MUFAFA (-), and PUFAFA (+). Robust relations (markers related to both MVPA and SED/prolSED) were found for a variety of VLDLs, serum TG, isoleucine, alanine, Gp, and FAw6FA (negative relations to MVPA and positive relations to SED/prolSED). Most separate relations were between HDLs and SED/prolSED, with a mix of markers separately related to MVPA.

ProlSED was related only to cholesterols in HDL (-) and MVPA was only related to cholesterols in VLDL (-). SED (+) and MVPA (-) but not prolSED were related to triglycerides in lipoproteins. SED and prolSED were negatively related to

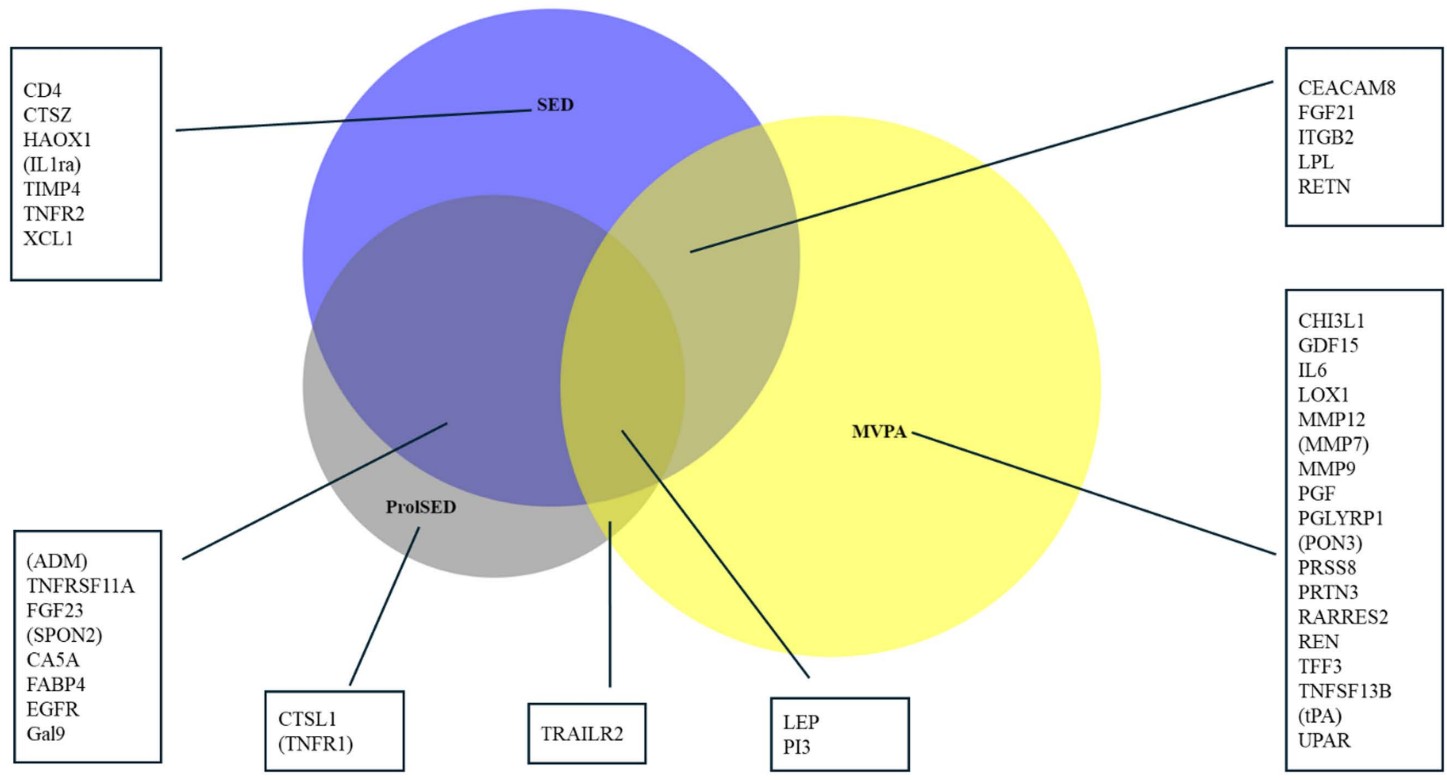

**Fig 2. Proportional Wenn-diagram for proteomic markers significantly related to the three PA classes, SED (light blue), proISED (dark blue), and MVPA (yellow).**

phospholipids in HDLs and positively related to phospholipids in VLDL. MVPA was only related to phospholipids in VLDLs (-). SED was related to total lipid content in HDL (-) and VLDL (+), with reversed associations for MVPA. MVPA was only related to total lipid content in VLDLs (-). Regarding particle size and concentration, SED was positively related to the size and concentration of VLDL but negatively related to HDLs. LIPA and MVPA were exclusively related to the size and concentration of VLDLs (-).

### Proteomic markers

Of the 43 related proteomic markers, a considerable proportion (18 markers) were uniquely related to MVPA. Only eight markers were jointly related to MVPA and SED and/or proISED which is significantly less than for the metabolic markers ($\chi2 = 22.5.0$, df = 1, p < 0.001). ProISED shared ten of thirteen markers with SED. Relations were nominally stronger for proteomic markers, compared to metabolomic markers (S4 Fig, Tables 2–4].

We found the nominally strongest associations for LEP (-), FGF21 (-), IL6 (-), MMP9 (-), and LOX1 (-), all associations with MVPA. (Tables 1 and 2) Proteomic markers significantly related to the PA classes were only weakly inter-related. Robust negative relations to MVPA (and positive to SED/proISED) were found for TRAILR2, LEP, PI3, CEACAM8, FGF21, and RETN. Opposite relations were found for ITGB2 and LPL, with positive relations to MVPA and negative to SED. Separate relations were noted for most significant markers, with exceptions for Il1ra, ADM, SPON2, TNFR1, MMP7, PON3, and tPA.

**Table 2. Markers significantly related to SED. Values are beta coefficients of the unadjusted model, model 1 and model 2.**

| Marker | Name | Beta unadjusted (SE) | Beta model 1 (SE) | Beta model 2 (SE) |
|---|---|---|---|---|
| | *Amino acids, fatty acids and ketones* | | | |
| Ala | Alanin | 0,000757398 (6,72444E-05) | 0,000683323 (6,9491E-05) | 0,000342272 (8,26063E-05) |
| Crea | Creatinine | 0,000206588 (1,48557E-05) | 0,000153736 (1,0683E-05) | 4,73282E-05 (8,96818E-06) |
| FAw6FA | Ratio of omega-6 fatty acids to total fatty acids | −0,035089296 (0,002763002) | −0,024238701(0,002738367) | −0,008927226 (0,003082267) |
| Gp | Glycoprotein acetyls, mainly a1-acid glycoprotein | 0,002082828 (0,000185562) | 0,001652642 (0,00018933) | 0,000458226 (0,000207641) |
| Ile | Iso-leucin | 0,000257519 (1,63193E-05) | 0,000149325 (1,49111E-05) | 6,04017E-05 (1,67903E-05) |
| Lac | Lactate | 0,00431094 (0,000380532) | 0,003250504 (0,000386103) | 0,001642492 (0,000463092) |
| Leu | Leucine | 0,000252418 (1,86517E-05) | 0,000120051 (1,65722E-05 | 4,36894E-05 (1,9182E-05) |
| MUFAFA | Ratio of monounsaturated fatty acids to total fatty acids | 0,035991143 (0,002801885) | 0,025614896 (0,002781239) | 0,010656412(0,003087197) |
| Phe | Phenylalanine | 7,02567E-05 (7,4994E-06) | 5,10294E-05 (7,7009E-06) | 2,02786E-05 (8,93771E-06) |
| Val | Valine | 0,000591608 (4,26133E-05) | 0,000331844 (3,97014E-05) | 0,000203482 (4,61558E-05) |
| PUFAFA | Ratio of polyunsaturated fatty acids to total fatty acids | −0,036229424 (0,003004184) | −0,023644925 (0,00181495) | −0,007859597 (0,003311037) |
| TGPG | Ratio of triglycerides to phosphoglycerides | 0,003697449 (0,000269492) | 0,002577079 (0,000269207) | 0,00112821(0,00030185) |
| UnSat | Estimated degree of unsaturation | −0,000598165 (5,92103E-05) | −0,000392529 (5,92365E-05) | −0,000160135 (6,83341E-05) |
| | *Apolipoproteins* | | | |
| ApoA1 | Apolipoprotein A-I | −0,003205023 (0,000223272) | −0,00178288 (0,001972295) | −0,001396951 (0,000244295) |
| ApoBApoA1 | ApoB-to-ApoA1 ratio | 0,002147062 (0,000173095) | 0,001544738 (0,000174665) | 0,000851516 (0,000204121) |
| | *Triglycerides in:* | | | |
| IDLTG | IDL | 0,000146643 (2,91207E-05) | 0,000170783 (3,04712E-05) | 7,76484E-05 (3,6111E-05 |
| LVLDLTG | large VLDL | 0,001485796 (0,00012601) | 0,001059108 (0,000127491) | 0,00035735 (0,000144793) |
| MVLDLTG | medium VLDL | 0,002201863 (0,000175888) | 0,00155024 (0,000177458) | 0,000602896 (0,000200989) |
| SHDLTG | small HDL | 0,000197954 (1,52525E-05) | 0,000134137 (1,51856E-05) | 5,47849E-05 (1,71227E-05) |
| SVLDLTG | small VLDL | 0,001309077 (0,000104074) | 0,000944677 (0,000105604) | 0,000408588 (0,000119909) |
| VLDLTG | VLDL | 0,005910212 (0,000484311) | 0,004250725 (0,000490816) | 0,001631061 (0,000557047) |
| XLVLDLTG | very large VLDL | 0,000440238 (4,01758E-05) | 0,000320525 (4,08346E-05) | 0,000100027 (4,68506E-05) |
| XSVLDLTG | very small VLDL | 0,000349484 (3,4231E-05) | 0,000285587 (3,55203E-05) | 0,000129653 (4,10133E-05) |
| XXLVLDLTG | chylomicrons and extremely large VLDL | 0,000170209 (1,68287E-05) | 0,000132453 (1,72914E-05) | 4,06475E-05 (2,00668E-05) |
| *SerumTG* | *Total serum TG* | 0,006288518 (0,000557214) | 0,0047758 (0,0005718) | 0,001795364 (0,000653029 |
| | *Cholesterols* | | | |
| HDL2C | Total cholesterol in HDL2 | −0,00595068 (0,000373095) | −0,003552335 (0,00034401) | −0,002380541 (0,00039767 |
| HDLC | Total cholesterol in HDL | −0,006319701 (0,000398226) | −0,003748321 (0,00036541 | −0,002518947 (0,00042423 |
| LHDLC | Total cholesterol in large HDL | −0,0033848 (0,000221823 | −0,001936516 (0,0002028) | −0,001168106 (0,00023176) |
| LHDLCE | Cholesterol esters in large HDL | −0,002575691 (0,000169224) | −0,001473052 (0,000154688) | −0,000895955 (0,000177116) |
| LHDLFC | Free cholesterol in large HDL | −0,000809016 (5,27052E-05) | −0,000463365 (4,81974E-05) | −0,000272034 (5,4795E-05) |
| LVLDLC | Total cholesterol in large VLDL | 0,0005744 (4,99384E-05) | 0,000425083 (5,09591E-05 | 0,000162113 (5,85212E-05) |
| LVLDLCE | Cholesterol esters in large VLDL | 0,000306376 (2,61113E-05) | 0,000229598 (2,66584E-05) | 9,80094E-05 (3,07937E-05) |
| LVLDLFC | Cholesterol esters in large VLDL | 0,000267983 (2,42264E-05) | 0,000195454 (2,46738E-05) | 6,40552E-05 (2,82497E-05) |
| MHDLC | Total cholesterol in medium HDL | −0,001237926 (9,41202E-05) | −0,000881011 (9,38655E-05) | −0,000821985 (0,000113623) |
| MHDLCE | Cholesterol esters in medium HDL | −0,000962191 (7,41617E-05) | −0,000692668(7,42398E-05) | −0,000648274 (8,9955E-05) |
| MHDLFC | Free cholesterol in medium HDL | −0,000275742 (2,04341E-05) | −0,000188349 (2,00818E-05) | −0,000173792 (2,42711E-05) |
| MVLDLC | Total cholesterol in medium VLDL | 0,000926256 (8,52754E-05) | 0,000725348 (8,77637E-05) | 0,000313733 (0,000102088) |

*(Continued)*

 

**Table 2.** (Continued)

| Marker | Name | Beta unadjusted (SE) | Beta model 1 (SE) | Beta model 2 (SE) |
|---|---|---|---|---|
| MVLDLCE | Cholesterol esters in medium VLDL | 0,000459677 (4,73061E-05) | 0,000381408 (4,89358E-05) | 0,000180353 (5,77761E-05) |
| MVLDLFC | Free cholesterol in medium VLDL | 0,000466661 (3,97015E-05) | 0,000344025 (4,05208E-05) | 0,00013344 (4,63021E-05) |
| SVLDLC | Total cholesterol in small VLDL | 0,000658636 (8,02733E-05) | 0,000562756 (8,34074E-05) | 0,000283848 (9,89433E-05) |
| SVLDLCE | Cholesterol esters in small VLDL | 0,000364147 (5,27335E-05) | 0,000328136 (5,48881E-05) | 0,000181352 (6,57135E-05) |
| SVLDLFC | Free cholesterol in small VLDL | 0,00029447 (2,99862E-05) | 0,00023458 (3,09752E-05) | 0,000102452 (3,60542E-05) |
| VLDLC | Total cholesterol in VLDL | 0,002411375 (0,000269418) | 0,002065524 (0,000279525) | 0,000945382 (0,000330269) |
| XLHDLFC | Free cholesterol in very large HDL | −0,000489711 (3,66678E-05) | −0,000237434 (3,28481E-05) | −0,000107204 (3,76589E-05) |
| XLVLDLCE | Cholesterol esters in very large VLDL | 7,88053E-05 (7,0647E-06) | 5,97828E-05 (7,23193E-06) | 2,4636E-05 (8,43465E-06) |
| XLVLDLFC | Free cholesterol in very large VLDL | 5,95526E-05 (5,84525E-06) | 4,56085E-05 (5,99609E-06) | 1,51758E-05 (6,97888E-06) |
| XLVLDLC | Total cholesterol in very large VLDL | 0,000138378 (1,28571E-05) | 0,000105409 (1,31899E-05) | 3,98348E-05 (1,53459E-05) |
| XXLVLDLC | Total cholesterol in chylomicrons and extremely large VLDL | 4,41523E-05 (4,38242E-06) | 3,56332E-05 (4,5269E-06) | 1,35055E-05 (5,30692E-06) |
| XXLVLDLCE | Cholesterol esters in chylomicrons and extremely large VLDL | 2,4846E-05 (2,56015E-06) | 2,09162E-05 (2,65561E-06) | 9,19626E-06 (3,14581E-06) |
| | *Phospholipids in:* | | | |
| LHDLPL | large HDL | −0,002800827 (0,000179897) | −0,001627117 (0,00016450) | −0,00106597 (0,00018976) |
| LVLDLPL | large VLDL | 0,000452934 (3,90739E-05) | 0,000328975 (3,96998E-05) | 0,000113296 (4,52355E-05) |
| MHDLPL | medium HDL | −0,000912957 (7,70612E-05) | −0,000613855 (7,6003E-05) | −0,000623675 (9,21013E-05) |
| MVLDLPL | medium VLDL | 0,00074992 (6,23607E-05) | 0,000547846 (6,34244E-05) | 0,000219399 (7,24465E-05) |
| SVLDLPL | small VLDL | 0,000493055 (4,73479E-05) | 0,000379489 (4,8694E-05) | 0,000161985 (5,61787E-05) |
| XLHDLPL | very large HDL | −0,002054681 (0,000146845) | −0,001028559 (0,00013092) | −0,000492023 (0,000149221) |
| XLVLDLPL | very large VLDL | 0,000111759 (1,07427E-05) | 8,40237E-05 (1,09902E-05) | 2,62132E-05 (1,26947E-05) |
| | *Total lipids in:* | | | |
| LHDLL | large HDL | −0,006342295 (0,000412701) | −0,003626729 (0,00037691) | −0,002280043 (0,00043196) |
| LVLDLL | large VLDL | 0,002513186 (0,000214448) | 0,001813235 (0,000217609) | 0,000632821 (0,000247801) |
| MHDLL | medium HDL | −0,002036793 (0,000171113) | −0,001406397 (0,00017000) | −0,001427366 (0,000206204) |
| MVLDLL | medium VLDL | 0,003878215 (0,000320133) | 0,002823594 (0,000325469) | 0,001136061 (0,000371424 |
| SVLDLL | small VLDL | 0,002460763 (0,000222008) | 0,001886902 (0,000228097) | 0,000854478 (0,000263458) |
| XLHDLL | very large HDL | −0,003635884 (0,000270317) | −0,001764195 (0,00024160) | −0,00082681 (0,000276791) |
| XLVLDLL | very large VLDL | 0,000690364 (6,34996E-05) | 0,000509953 (6,47649E-05) | 0,000165993 (7,45401E-05) |
| XXLVLDLL | chylomicrons and extremely large VLDL | 0,000244571 (2,41465E-05) | 0,000191598 (2,48328E-05) | 6,06016E-05 (2,8848E-05) |
| | *Particle sizes:* | | | |
| HDLD | Mean diameter for HDL particles | −0,005039151 (0,00035135) | −0,002550716 (0,00031293) | −0,001244384 (0,000354833) |
| LHDLP | Concentration of large HDL particles | −9,87364E-09 (6,43413E-10) | −5,635e-09 (5,86774E-1 | −3,56704E-09 (6,73749E-10) |
| LVLDLP | Concentration of large VLDL particles | 4,34405E-11 (3,69497E-12) | 3,12621E-11 (3,7477E-12) | 1,08648E-11 (4,26369E-12) |
| MHDLP | Concentration of medium HDL particles | −4,52376E-09 (3,93504E-10) | −3,11244E-09 (3,9139E-10) | −3,23756E-09 (4,74873E-10) |
| MVLDLP | Concentration of medium VLDL particles | 1,17455E-10 (9,61577E-12) | 8,49667E-11 (9,76432E-12) | 3,40594E-11 (1,11231E-11) |
| SVLDLP | Concentration of small VLDL particles | 1,31344E-10 (1,15039E-11) | 9,95346E-11 (1,18104E-11) | 4,47564E-11 (1,35769E-11) |
| VLDLD | Mean diameter for VLDL particles | 0,021243423 (0,001507626) | 0,014063394 (0,001475238) | 0,006080771 (0,001662287) |
| XLHDLP | Concentration of very large HDL particles | −3,57524E-09 (2,65935E-10) | −1,73154E-09 (2,3756E-10) | −8,16207E-10 (2,72231E-10) |
| XLVLDLP | Concentration of very large VLDL particles | 7,0986E-12 (6,51041E-13) | 5,22434E-12 (6,63501E-13) | 1,68662E-12 (7,62916E- |
| XSVLDLP | Concentration of very small VLDL particles | 4,40861E-11 (1,0309E-11) | 5,3469E-11 (1,08305E-11) | 2,656E-11 (1,29434E-11) |
| XXLVLDLP | Concentration of chylomicrons and extremely large VLDL particles | 1,14038E-12 (1,12585E-13) | 8,92743E-13 (1,15748E-13) | 2,8134E-13 (1,34462E-13) |
| | *Proteins* | | | |

*(Continued)*

**Table 2.** (Continued)

| Marker | Name | Beta unadjusted (SE) | Beta model 1 (SE) | Beta model 2 (SE) |
|--------|------|----------------------|-------------------|-------------------|
| ADM | Pro-adrenomedullin | 0,007446295 (0,000534891) | 0,007359451 (0,000551924) | 0,003457726 (0,000596551) |
| CA5A | Carbonic anhydrase 5A | 0,011168224 (0,001126814) | 0,007207642 (0,001156878) | 0,002721892 (0,00136308) |
| CD4 | T-cell surface glycoprotein CD4 | 0,002691825 (0,000389497) | 0,00284425 (0,000415379) | 0,001185998 (0,000496927) |
| CEACAM8 | Carcinoembryonic antigen-related cell adhesion molecule 8 | 0,005953472 (0,000607435) | 0,0046655 (0,00063822) | 0,001833103 (0,000762638) |
| CTSZ | Cathepsin Z | 0,005115518 (0,000535522) | 0,003615147 (0,000554227) | 0,001567717 (0,000649252) |
| EGFR | Epidermal growth factor receptor | −0,000891032 (0,00027802) | −0,000733592(0,00031096) | −0,001065195 (0,000362402) |
| FABP4 | Fatty acid-binding protein, adipocyte | 0,00756275 (0,000987665) | 0,01119042 (0,000908023) | 0,00357126 (0,000939422) |
| FGF21 | Fibroblast growth factor 21 | 0,019073166 (0,001881712) | 0,017273155 (0,001953605) | 0,006661382 (0,002235862) |
| FGF23 | Fibroblast growth factor 23 | 0,004162749 (0,000605142) | 0,004631573 (0,000642545) | 0,002362016 (0,000765342) |
| Gal9 | Galectin-9 | 0,003435648 (0,000448696) | 0,003281429 (0,000461537) | 0,001481964 (0,000531889) |
| HAOX1 | Hydroxyacid oxidase 1 | 0,015888091 (0,001733985) | 0,009674681 (0,001759726) | 0,004553619 (0,002101628) |
| IL1ra | Interleukin-1 receptor antagonist protein | 0,010546778 (0,000967533) | 0,0097498 (0,000950495) | 0,003236482 (0,001030126) |
| ITGB2 | Integrin beta-2 | −0,003947141 (0,000491539) | −0,00358516 (0,000514364) | −0,003177349 (0,000619599) |
| LEP | Leptin | 0,012177215 (0,001604368) | 0,02013132 (0,001254941) | 0,007419023 (0,001241799) |
| LPL | Lipoprotein lipase | −0,010698792 (0,000630343) | −0,007096573 (0,00057841) | −0,0055346 (0,000703881) |
| PI3 | Elafin | 0,007259485 (0,00069993) | 0,005636906 (0,000724181) | 0,002456884 (0,000860147) |
| RETN | Resistin | 0,004342793 (0,000590814) | 0,004665533 (0,000630287) | 0,002395942(0,000750952) |
| SPON2 | Spondin-2 | 0,001584103 (0,000240082) | 0,001654546 (0,000253891) | 0,000671623 (0,000301951) |
| TIMP4 | Metalloproteinase inhibitor 4 | 0,001086715 (0,00056827) | 0,002920281 (0,000582809) | 0,001466391 (0,000693856) |
| TNFR1 | Tumor necrosis factor receptor 1 | 0,005565843 (0,000452946) | 0,004834209 (0,000471281) | 0,002065514 (0,000538235) |
| TNFR2 | Tumor necrosis factor receptor 2 | 0,004747785 (0,000510376) | 0,004020958 (0,000527799) | 0,001919902 (0,000625728) |
| TNFRSF11A | Tumor necrosis factor receptor superfamily member 11A | 0,005346519 (0,000513226) | 0,005409847 (0,000541223) | 0,002900348 (0,000620613) |
| XCL1 | Lymphotactin | 0,002613806 (0,000766483) | 0,003301193 (0,000827666) | 0,001935281 (0,000985466) |

**Non-linear associations.** Potential non-linearity associations between significant biomarkers and PA classes were explored using a second-degree term. For MVPA, only three of the 103 significant markers showed a significant second-degree term, which is close to the expected number of findings. For SED, forty-seven of the 127 significant biomarkers showed a significant second-degree association, which is more than expected. However, explanatory values of the non-linear associations were not significantly higher than linear, and visual inspection did not reveal any meaningful cut-off points for the associations.

**Interaction between PA class and sex.** The interaction between PA class and sex in the significant associations was further investigated.

For SED, associations were stronger in men for the metabolomic markers FAw6FA, Ile, and Leu and a group of VLDL--related markers, while HDL-related markers generally showed a stronger relation to SED in women. Associations between SED and proteomic markers RETN and SPON2 were both strongly related in men. For SED, relations were in the same direction for both sexes. For MVPA, five HDL-related markers and proteomic markers ITGB2 were all strongly related in females. Interestingly, for MVPA, relations were opposite for men and women except for ITGB2. (S2 and S3 Tables).

**Interaction between PA class and waist circumference.** For SED, associations to VLDL-related markers and UnSat were significantly stronger among participants with low waist circumference, while creatinine, lactate, phenylalanine, and proteomic markers FAPB4, HAOX1, PARP1, SPON2, TIMP4, and TNFRSF11 showed stronger associations to SED in participants with high waist circumference. All relations between SED and biomarkers were in the same direction for Low and High WC.

**Table 3. Markers significantly related to proISED. Values are beta coefficients from unadjusted model, model 1, and model 2.**

| Marker | Name | Beta unadjusted (SE) | Beta model 1 (SE) | Beta model 2 (SE) |
|---|---|---|---|---|
| | *Amino acids, fatty acids and ketones* | | | |
| Ala | Alanin | 5,75413E-05 (7,13547E-06) | 4,72956E-05 (7,61497E-06) | 1,70196E-05 (7,97096E-06) |
| Crea | Creatinine | 1,62948E-05 (1,57815E-06) | 1,2152E-05 (1,16368E-06) | 3,29834E-06 (8,65408E-07) |
| Lac | Lactate | 0,000416154 (4,02922E-05) | 0,000304941 (4,23551E-05) | 0,000187495 (4,46365E-05) |
| Phe | Phenylalanine | 6,80737E-06 (7,93749E-07) | 4,71088E-06 (8,46153E-07) | 2,38666E-06 (8,61623E-07) |
| | *Apolipoprotein:* | | | |
| ApoA1 | Apolipoprotein A-I | −0,000248806 (2,37354E-05) | −0,000129877 (2,24717E-05) | −6,98032E-05 (2,35894E-05) |
| | *Cholesterol and cholesterol esters:* | | | |
| HDL2C | Total cholesterol in HDL2 | −0,000453316 (3,97217E-05) | −0,000249333 (3,51116E-05) | −0,000106809 (3,84093E-05) |
| HDLC | Total cholesterol in HDL | −0,00048184 (4,23939E-05) | −0,000263807 (3,99499E-05) | −0,000113594 (4,09741E-05) |
| LHDLC | Total cholesterol in large HDL | −0,000256243 (2,3605E-05) | −0,000130745 (2,22266E-05) | −4,54449E-05 (2,23754E-05) |
| LHDLCE | Cholesterol esters in large HDL | −0,000195458 (1,80066E-05) | −0,000100041 (1,58769E-05) | −3,5466E-05 (1,70997E-05) |
| MHDLC | Total cholesterol in medium HDL | −0,000103954 (9,98879E-06) | −7,56782E-05 (1,02903E-05) | −5,46403E-05 (1,0974E-05) |
| MHDLCE | Cholesterol esters in medium HDL | −8,15207E-05 (7,86909E-06) | −6,01198E-05 (8,15666E-06) | −4,38739E-05 (8,68748E-06) |
| MHDLFC | Free cholesterol in medium HDL | −2,24368E-05 (2,16985E-06) | −1,55625E-05 (2,202E-06) | −1,07764E-05 (2,34454E-06) |
| SVLDLCE | Cholesterol esters in small VLDL | 3,03649E-05 (5,58258E-06) | 2,66092E-05 (6,05147E-06) | 1,28808E-05 (6,33734E-06) |
| | *Phospholipids in:* | | | |
| LHDLPL | large HDL | −0,000213246 (1,91468E-05) | −0,000112972 (1,80184E-05) | −4,53817E-05 (1,83251E-05) |
| MHDLPL | medium HDL | −7,70391E-05 (8,1726E-06) | −5,42839E-05 (7,80536E-06) | −4,16509E-05 (8,89338E-06) |
| | *Total lipids in:* | | | |
| LHDLL | large HDL | −0,000481475 (4,39193E-05) | −0,00024765 (4,12621E-05) | −9,21692E-05 (4,17084E-05) |
| MHDLL | medium HDL | −0,000174058 (1,81454E-05) | −0,000125655 (1,87233E-05) | −9,7655E-05 (1,99112E-05) |
| | *Particle sizes:* | | | |
| LHDLP | Concentration of large HDL particles | −7,50161E-10 (6,84696E-11) | −3,85555E-10 (6,42494E-11) | −1,45017E-10 (6,50543E-11) |
| MHDLP | Concentration of medium HDL particles | −3,89794E-10 (4,17176E-11) | −2,82222E-10 (4,29098E-11) | −2,24501E-10 (4,58501E-11) |
| | *Proteins:* | | | |
| ADM | Pro-adrenomedullin | 0,000647433 (5,67638E-05) | 0,000593559 (6,08141E-05) | 0,000291703 (5,85873E-05) |
| CA5A | Carbonic anhydrase 5A, mitochondrial | 0,000957928 (0,00011939) | 0,00061161 (0,00013099) | 0,000289831 (0,000133788) |
| CTSL1 | Cathepsin L1 | 0,000253652 (4,62612E-05) | 0,000160632 (4,65883E-05) | 0,000116998 (5,24017E-05) |
| EGFR | Epidermal growth factor receptor | −8,85146E-05 (2,94056E-05) | −6,90074E-05 (3,4407E-05) | −8,98081E-05 (3,55784E-05) |
| FABP4 | Fatty acid-binding protein, adipocyte | 0,000660856 (0,000104561) | 0,000896701 (9,49827E-05) | 0,00035056 (9,22136E-05) |
| FGF23 | Fibroblast growth factor 23 | 0,000328836 (6,40726E-05) | 0,000365183 (7,24377E-05) | 0,000190799 (7,51372E-05) |
| Gal9 | Galectin-9 | 0,000349071 (4,74661E-05) | 0,000296887 (5,07129E-05) | 0,000171524 (5,12729E-05) |
| LEP | Leptin | 0,000733019 (0,000170036) | 0,001361011 (0,000132829) | 0,000313281 (0,000122112) |
| PI3 | Elafin | 0,000635571 (7,60459E-05) | 0,000473655 (8,26773E-05) | 0,000237512 (8,65987E-05) |
| SPON2 | Spondin-2 | 0,000173148 (2,53873E-05) | 0,000170664 (2,83772E-05) | 0,00010115 (2,96257E-05) |
| TNFR1 | Tumor necrosis factor receptor 1 | 0,00050181 (4,80113E-05) | 0,000406556 (5,12016E-05) | 0,000189236 (5,19002E-05) |
| TNFRSF11A | Tumor necrosis factor receptor super-family member 11A | 0,000550731 (5,42958E-05) | 0,000540855 (5,624E-05) | 0,000325564 (6,0891E-05) |
| TRAILR2 | TNF-related apoptosis-inducing ligand receptor 2 | 0,000336565 (5,06483E-05) | 0,00026247 (5,47382E-05) | 0,000125594 (5,29428E-05) |

**Table 4. Markers significantly related to MVPA. Values are beta coefficients from unadjusted model, model 1, and model 2.**

| Marker | Name | Beta unadjusted (SE) | Beta model 1 (SE) | Beta model 2 (SE) |
|---|---|---|---|---|
| | *Amino acids, fatty acids and ketones* | | | |
| Ala | Alanine | −0,001997658 (0,000204386) | −0,001811386 (0,00020704) | −0,00085764 (0,000243055) |
| Gp | Glycoprotein acetyls, mainly a1-acid glycoprotein | −0,006675625 (0,000562613) | −0,006158627 (0,00057301) | −0,002791355 (0,000611732) |
| Ile | Isoleucine | −0,000312563 (5,00785E-05) | −0,00037182 (4,44112E-05) | −0,000119901 (4,93956E-05) |
| bOHBut | 3-hydroxybutyrate | 0,001344012 (0,000506077) | 0,001222472 (0,00050869) | 0,001257048 (0,000627057) |
| FAw6FA | Ratio of omega-6 fatty acids to total fatty acids | 0,091508589 (0,008403092) | 0,085137226 (0,008236262) | 0,030175549 (0,009073206) |
| LAFA | Ratio of 18:2 linoleic acid to total fatty acids | 0,082977986 (0,008521324) | 0,072476319 (0,008283828) | 0,027468151 (0,009518422) |
| TGPG | Ratio of triglycerides to phosphoglycerides | −0,008027147 (0,000821807) | −0,008207828 (0,00081024) | −0,003131338 (0,00088834) |
| PUFAFA | Ratio of polyunsaturated fatty acids to total fatty acids | 0,093283636 (0,009135865) | 0,088563088 (0,008891729) | 0,029655329 (0,009747205) |
| MUFA | Monounsaturated fatty acids | −0,020985929 (0,002461154) | −0,0193762 (0,002534124) | −0,007706282 (0,002885652) |
| MUFAFA | Ratio of monounsaturated fatty acids to total fatty acids | −0,093028207 (0,00852279) | −0,087726216 (0,00840271) | −0,031206272 (0,009086148) |
| | *Triglycerides in:* | | | |
| HDLTG | HDL | −0,000848775 (0,000113039) | −0,000780756 (0,00011654) | −0,000477207 (0,000137427) |
| MHDLTG | medium HDL | −0,0003581 (4,49834E-05) | −0,000341671 (4,57326E-05) | −0,000135641 (5,20453E-05) |
| SHDLTG | small HDL | −0,000419474 (4,64978E-05) | −0,000424427 (4,55902E-05) | −0,000139327 (5,03875E-05) |
| SLDLTG | small LDL | −0,000185789 (2,70266E-05) | −0,000171566 (2,80061E-05) | −6,62381E-05 (3,25961E-05) |
| LVLDLTG | large VLDL | −0,003653189 (0,00038336) | −0,00368966 (0,000384635) | −0,001498335 (0,000426318) |
| MVLDLTG | medium VLDL | −0,004935679 (0,000535763) | −0,005102922 (0,00053405) | −0,00194138 (0,000591614) |
| SVLDLTG | very small VLDL | −0,002874237 (0,000317085) | −0,00296683 (0,00031701) | −0,001052928 (0,000352857) |
| VLDLTG | medium VLDL | −0,013780726 (0,001474422) | −0,014057152 (0,00147675) | −0,00537104 (0,001639716) |
| XLVLDLTG | chylomicrons and extremely large VLDL | −0,001137596 (0,000122125) | −0,001139697 (0,00012276) | −0,000487657 (0,000137958) |
| XSVLDLTG | x-small VLDL | −0,000872787 (0,000104058) | −0,000852277 (0,00010626) | −0,000272007 (0,000120673) |
| XXLVLDLTG | chylomicrons and extremely large VLDL | −0,000464492 (5,11182E-05) | −0,000465627 (5,17935E-05) | −0,000218052 (5,90954E-05) |
| *SerumTG* | Serum total triglycerides | −0,015931527 (0,001694315) | −0,015926478 (0,00171542) | −0,006221542 (0,001922339) |
| | *Cholesterol and cholesterol esters:* | | | |
| LVLDLC | Total cholesterol in large VLDL | −0,001370512 (0,000151945) | −0,001409557 (0,00015267) | −0,000569627 (0,000172274) |
| LVLDLCE | Cholesterol esters in large VLDL | −0,000689417 (7,94966E-05) | −0,000725759 (7,97544E-05) | −0,000286853 (9,06312E-05) |
| LVLDLFC | Free cholesterol in large VLDL | −0,000681152 (7,36567E-05) | −0,000683847 (7,40348E-05) | −0,000282875 (8,31775E-05) |
| MVLDLC | Total cholesterol in medium VLDL | −0,0021311 (0,000259457) | −0,002239775 (0,00026243) | −0,000880053 (0,00030045) |
| MVLDLCE | Cholesterol esters in medium VLDL | −0,001029626 (0,00014388) | −0,001108994 (0,00014683) | −0,000435955 (0,000170013) |
| MVLDLFC | Free cholesterol in medium VLDL | −0,001101776 (0,000120824) | −0,001131103 (0,00012161) | −0,000444413 (0,000136296) |
| SVLDLFC | Free cholesterol in small VLDL | −0,000668989 (9,1197E-05) | −0,000681991 (9,30445E-05) | −0,000208491 (0,000106087) |
| VLDLC | Total cholesterol in large VLDL | −0,005571649 (0,000818992) | −0,005808698 (0,00083929) | −0,002029188 (0,000971815) |
| XLVLDLC | Total cholesterol in very large VLDL | −0,000338969 (3,90982E-05) | −0,000349556 (3,94117E-05) | −0,00015193 (4,51787E-05) |
| XLVLDLCE | Cholesterol esters in very large VLDL | −0,000183233 (2,14963E-05) | −0,000192334 (2,1626E-05) | −8,22527E-05 (2,48286E-05) |
| XLVLDLFC | Free cholesterol in very large VLDL | −0,000155752 (1,77621E-05) | −0,000157235 (1,79563E-05) | −6,9732E-05 (2,05488E-05) |
| XXLVLDLC | Total cholesterol in chylomicrons and extremely large VLDL | −0,0001157 (1,33163E-05) | −0,000118391 (1,35317E-05) | −5,48797E-05 (1,5625E-05) |
| XXLVLDLCE | Cholesterol esters in chylomicrons and extremely large VLDL | −6,18548E-05 (7,78131E-06) | −6,52311E-05 (7,94284E-06) | −3,05962E-05 (9,2601E-06) |
| XXLVLDLFC | Free cholesterol in chylomicrons and extremely large VLDL | −5,38781E-05 (5,81818E-06) | −5,31954E-05 (5,88644E-06) | −2,43219E-05 (6,69699E-06) |

*(Continued)*

| Marker | Name | Beta unadjusted (SE) | Beta model 1 (SE) | Beta model 2 (SE) |
|---|---|---|---|---|
| | *Phospholipids in:* | | | |
| LVLDLPL | large VLDL | −0,001115886 (0,000118858) | −0,001127558 (0,00011949) | −0,000455741 (0,000133182) |
| MVLDLPL | medium VLDL | −0,001687538 (0,000189891) | −0,001753362 (0,00018996) | −0,000663592 (0,000213231) |
| SVLDLPL | small VLDL | −0,00109916 (0,000144058) | −0,001118583 (0,00014611) | −0,000324217 (0,000165299) |
| XLVLDLPL | very large VLDL | −0,000294114 (3,26445E-05) | −0,000294821 (3,20144E-05) | −0,00012784 (3,73803E-05) |
| XXLVLDLPL | chylomicrons and extremely large VLDL | −8,62525E-05 (9,37464E-06) | −8,49089E-05 (9,50438E-06) | −3,89754E-05 (1,08257E-05) |
| | *Total lipids in:* | | | |
| LVLDLL | large VLDL | −0,006139584 (0,000652414) | −0,006226773 (0,00065472) | −0,002523624 (0,000729571) |
| MVLDLL | medium VLDL | −0,008754832 (0,000974844) | −0,009096542(0,000977114) | −0,003485526 (0,001093228) |
| SVLDLL | small VLDL | −0,005274101 (0,000675893) | −0,005491108 (0,00068294) | −0,001754483 (0,000775147) |
| XLVLDLL | very large VLDL | −0,001770861 (0,000193032) | −0,001784267 (0,00019406) | −0,000767744 (0,000219485) |
| XXLVLDLL | chylomicrons and extremely large VLDL | −0,000666425 (7,33475E-05) | −0,000668905 (7,4405E-05) | −0,000311885 (8,49529E-05) |
| | *Particle size:* | | | |
| LVLDLP | Concentration of large VLDL particles | −1,06107E-10 (1,12414E-11) | −1,07565E-10 (1,12881E-11) | −4,35723E-11 (1,25532E-11) |
| MVLDLP | Concentration of medium VLDL particles | −2,64211E-10 (2,92838E-11) | −2,74424E-10 (2,93226E-11) | −1,04815E-10 (3,27391E-11) |
| SVLDLP | Concentration of small VLDL particles | −2,82217E-10 (3,50296E-11) | −2,93511E-10 (3,52522E-11) | −9,5601E-11 (3,99468E-11) |
| VLDLD | Mean diameter for VLDL particles | −0,04492516 (0,004599996) | −0,04635674 (0,004445012) | −0,018100394 (0,004892523) |
| XLVLDLP | Concentration of very large VLDL particles | −1,82256E-11 (1,9791E-12) | −1,83399E-11 (1,98932E-12) | −7,86986E-12 (2,24644E-12) |
| XXLVLDLP | Concentration of chylomicrons and extremely large VLDL particles | −3,10609E-12 (3,4199E-13) | −3,11753E-12 (3,46775E-13) | −1,45393E-12 (3,95971E-13) |
| | *Proteins* | | | |
| CEACAM8 | Carcinoembryonic antigen-related cell adhesion molecule 8 | −0,013298882 (0,001847549) | −0,014556686 (0,00191405) | −0,009631684 (0,002241865) |
| CHI3L1 | Chitinase-3-like protein 1 | −0,022971991 (0,003384647) | −0,01859018 (0,003508857) | −0,008647311 (0,004045689) |
| FGF21 | Fibroblast growth factor 21 | −0,070053334 (0,005694919) | −0,061600316 (0,00590028) | −0,028249103 (0,006571312) |
| GDF15 | Growth/differentiation factor 15 | −0,017464513 (0,002007169) | −0,013460106 (0,00194742) | −0,007778152 (0,002181139) |
| IL6 | Interleukin-6 | −0,033176711 (0,003011543) | −0,02826306 (0,003056445) | −0,016723585 (0,003485255) |
| ITGB2 | Integrin beta-2 | 0,010231473 (0,001492903) | 0,013163923 (0,001536924) | 0,008061274 (0,00181918) |
| LEP | Leptin | −0,071021878 (0,00482695) | −0,052075226 (0,00368386) | −0,019940734 (0,003646182) |
| LOX1 | Lipoxygenase-1 | −0,018245904 (0,001960759) | −0,017345066 (0,00200973) | −0,012978217 (0,00237881) |
| LPL | Lipoproteinlinase | 0,012131768 (0,001938325) | 0,014898467 (0,001785016) | 0,004199866 (0,00206001) |
| MMP7 | Matrilysin | −0,013531344 (0,001502548) | −0,011039544 (0,00155021) | −0,007326018 (0,00180792) |
| MMP9 | Matrix metalloproteinase-9 | −0,022860288 (0,003071323) | −0,023206698 (0,00316118) | −0,0165899 (0,003705674) |
| MMP12 | Matrix metalloproteinase-12 | −0,018847613 (0,002510292) | −0,014898427 (0,00250909) | −0,009747345 (0,003043283) |
| PGF | Placenta growth factor | −0,006094464 (0,001096434) | −0,006663989 (0,00110869) | −0,003465778 (0,00126799) |
| PGLYRP1 | Peptidoglycan recognition protein 1 | −0,010924482 (0,001902916) | −0,011527805 (0,00200254) | −0,005172609 (0,002333144) |
| PI3 | Elafin | −0,015131332 (0,002144302) | −0,018128302 (0,00218204) | −0,009887833 (0,002540709) |
| PON3 | Paraoxonase | 0,021626673 (0,002758333) | 0,018190876 (0,00261885) | 0,006281143 (0,002927796) |
| PRSS8 | Prostasin | −0,013653318 (0,001724929) | −0,012535991 (0,0016866) | −0,006068555 (0,001883755) |
| PRTN3 | Myeloblastin | −0,009462432 (0,001856551) | −0,009624614 (0,00197347) | −0,004607934 (0,002316056) |
| RARRES2 | Retinoic acid receptor responder protein 2 | −0,015754495 (0,001468753) | −0,013622475 (0,00149527) | −0,005501454 (0,001632155) |
| REN | Renin | −0,018404205 (0,003362319) | −0,018367703 (0,00334094) | −0,011326584 (0,003892206) |
| RETN | Resistin | −0,01228823 (0,001793415) | −0,01243717 (0,001894795) | −0,006682177 (0,002205628) |
| TFF3 | Trefoil factor 3 | −0,011514194 (0,001611293) | −0,010814729 (0,00165611) | −0,004751389 (0,001933227) |
| TNFSF13B | Tumor necrosis factor ligand superfamily member 13B | −0,010879679 (0,001430291) | −0,008632136 (0,00142081) | −0,004814957 (0,00173084) |

*(Continued)*

**Table 4.** (Continued)

| Marker | Name | Beta unadjusted (SE) | Beta model 1 (SE) | Beta model 2 (SE) |
|--------|------|---------------------|-------------------|-------------------|
| tPA | Tissue-type plasminogen activator | −0,024025346 (0,003029865) | −0,021732233 (0,00301566) | −0,010129752 (0,003515359) |
| TRAILR2 | TNF-related apoptosis-inducing ligand receptor 2 | −0,012419908 (0,001450857) | −0,010752076 (0,00147532) | −0,004320619 (0,001617029) |
| UPAR | Urokinase plasminogen activator surface receptor | −0,012919005 (0,001428267) | −0,010474066 (0,00140669) | −0,006006859 (0,001647695) |

ApoB-to-ApoA1 and three VLDL-related markers were strongly related to MVPA in participants with low waist circumference, while UnSat was strongly related to MVPA among individuals with high waist circumference. Two proteomic markers (CTSZ and HAOX1) showed significantly stronger associations with MVPA in participants with high waist circumference, while LEP was strongly related to MVPA in participants with low waist circumference. Except for UnSat and CTSZ, relations between biomarkers and MVPA were in the opposite direction for Low and High WC (S4 and S5 Tables).

## Discussion

To our knowledge, this is the first time that the association between accelerometer-based measures of PA at different intensities and plasma metabolomic and proteomic biomarkers has been investigated in two population-based samples. Using two independent samples for discovery and validation, we identified significant associations between several metabolomic and proteomic markers and the investigated PA classes.

We found associations between PA and eighty-five metabolomic species identified by metabolomic profiling and forty-three cardiovascular biomarkers from proteomic profiling in these two independent population-based cohorts. Metabolic markers were to a larger degree related to several PA classes while proteomic markers were significantly more often uniquely related to one PA class. MVPA, i.e., PA of higher intensity showed the strongest association with markers, both metabolic and proteomic. Although speculative, this may indicate if repeated in experimental trials, that this PA class may have a stronger association with CVD risk compared to SED or prolSED.

Prolonged SED was only marginally different from SED for metabolic markers but exhibited unique associations with several proteomic biomarkers. LIPA almost totally shared markers with SED. This makes it questionable if one of these two PA classes adds any meaningful information in risk assessment in middle-aged adults in the presence of the other.

It may be that SED lacks additional information to LIPA and that the relations between PA and biomarkers may be intensity dependent. In such interpretation, PA at higher intensities (e.g., MVPA) has the strongest relationship to biomarkers, and LIPA has a weaker relationship. This is supported by the findings that the strongest relationships in both metabolic and proteomic markers were found for MVPA.

As proteomic markers, rather than metabolic markers, distinguished the SED and MVPA PA classes, they may be central to explaining varying associations between different PA classes and CVD health. In longitudinal epidemiological studies, SED has been related to CVD mortality primarily in individuals lacking MVPA [27]. Proteomic markers uniquely related to MVPA may therefore be of interest when identifying the reason behind this. In a recent paper by Siegbahn et al [28], a novel CVD risk panel consisting of twenty-one biomarkers was validated. Among the most prognostic markers were MMP-12, UPAR, REN, and TFF3 (all related to MVPA in the present data) as well as ADM and FGF23 (both related to SED). This could indicate that these markers are of special importance to explain the above findings.

### Metabolomic profiling

LAFA (or linoleic fatty acids) [29], FAw6FA [30]and PUFAFA [31] are related to cardio-protective effects via different pathways and were found to be related to MVPA. Their positive associations with MVPA may be linked to the observed associations between MVPA and CVD. MUFAFA has been linked to an increased risk for CVD [32].

As significantly more metabolomic markers were jointly related to SED and prolSED, the proposed unique association between prolSED and vascular health [33], may be related to metabolic factors, rather than proteomic. Experimental work could be applied to verify this and examine any causal links.

In early work, Kujala et al [34] showed similar associations to ours between PA and metabolomic CVD markers. The analyses were partly conducted in twin pairs and indicate that levels of these biomarkers are, to a limited degree, genetically dependent since they differ between active and inactive twins. Interestingly, this included a1-acid glycoprotein, an inflammatory marker related to CVD risk. In the present study, we observed associations between glycoprotein acetyls and SED/LIPA and MVPA. Glycoprotein acetyls are not related to SED in young individuals [35], which could play a role in the differing associations between time spent on SED and metabolic risk markers among young age and adults.

SED/LIPA and prolSED, alone and in combination were negatively related to medium, large, and very large HDLs. MVPA was related to a limited number of HDL types. The majority of the metabolomic markers were jointly related to both MVPA and SED/LIPA. Those markers were predominantly LDLs or VLDLs. If causal, these findings may be of importance when prescribing PA to patients with blood lipid disorders.

In a sample of adults at high risk for type 2 diabetes, Henson et al. [36] showed negative associations between medium and small HDLs and PA corresponding to SED and positive associations between PA corresponding to MVPA and large, medium, and small-sized HDL. Further, they showed positive associations between LDLs and PA corresponding to SED, but no association to higher intensities. In their study, the intensity corresponding to SED was negatively related to ApoA1 but unrelated to the ApoB-to-ApoA1 ratio. We could not fully confirm these findings, as we found associations between SED and both ApoA1 and the ApoB-to-ApoA1 ratio in this mixed population. Based on data from younger men (27 yrs.), Vaara et al [37] reported strong, significant associations between ApoB-to-ApoB1-ratio and both SED and MVPA, indicating that these associations may be stronger in younger or more healthy populations. Differences between studies may be dependent on the PA definition and partly dependent on the different models used and the populations studied.

## Proteomic profiling

We found the strongest robust associations between MVPA and LEP (-), FGF21 (-), IL6 (-), MMP9 (-) and LOX1 (-). Leptin is related to cardiovascular dysfunction [38,39] and previous research has shown that LEP may decrease after exercise training [40]. IL6, MMP9 [41], and LOX1 [42] have been linked to inflammation, insulin resistance, or arteriosclerosis. Both MMP9 [43] and LOX1 [44] are further linked to a higher risk of ruptures of atherosclerotic plaques. Previous research has linked LOX1 to endothelial oxidated-LDL uptake, and thereby endothelial dysfunction.

A group of robustly related markers has been linked to inflammation or immune response markers, including TNFRSF11A [45], and Gal9 [46]. Inflammatory markers separately related to MVPA were IL-6 [47], MMP9 [48], MMP12 [49], CHI3L1 [50], TNFSF13B [50], TFF3 [51], and PGF [52]. Further, FGF23 [53], and GDF-15 [54], were robustly related to PA and have been linked to metabolic disorders such as diabetes or obesity. Renin, GDF15 [55], PRSS8 [56], RAR-RES2 [57], PGLYRP1 [58], LOX1 [59], and UPAR [60], all robustly related to PA, have been related to cardiovascular or kidney disease via different pathways.

In a recent paper, Lind et al. [61] found forty-two proteomic markers linked to incident CVD, of which eleven were related to MVPA in the present study and nine others to SED, with only two (FGF-21 and RETN) shared across PA classes. In the paper by Lind et al., forty-nine proteins were linked to MI, of which ten were related to SED and ten to MVPA, with only FGF-21 being related to both PA classes. Further, in a study of over 800 Olink proteins [62], 19 different markers were found to be predictive of future myocardial infarction. Among these were TRAILR2. MMP-12, GDF-15, CHI3L1, and IL-6 all were related to MVPA in the present analyses, and TRAILR2 was also related to prolSED.

However, there may be site-specific associations between PA classes and arteriosclerosis, with different markers indicating specific mechanisms. In previous work on SCAPIS data, time spent in SED was found to be related to a lower prevalence of carotid atherosclerosis but a risk factor for coronary stenosis [63]. Based on results from

the present investigation, biomarkers separately related to SED may therefore be candidates for explaining these opposite associations. For example, TNFR2 which is positively related to SED in the present paper, is considered a risk factor for myocardial infarction [64], but earlier studies have identified TNFR2 signaling as protective after ischemic strokes [65,66]. Another example is LPL, which is negatively related to SED and considered to be protective for coronary artery disease [67], but some genetic variants of the LPL gene have been related to a higher risk of carotid stenosis [68].

While relations to PA classes might share physiological mechanisms, there is most probably a large number of different pathways involved in the identified relationships between PA classes and biomarkers. For example, regarding metabolomic markers, PA is known to increase lipid metabolism and could thereby affect lipid profile via LPL activation or (up-)regulation of LDL receptors. Another example regarding proteomic markers related to inflammation (such as IL-6) may be affected by PA by alterations to the autonomic system, AMPK activation or changes in for example visceral fat deposits.

### Interactions

Associations between SED or MVPA and HDL-related markers were stronger in females, while associations with VLDL markers were stronger in males. Sex differences in associations between PA and some proteomic markers were also noted. Several of these proteomic markers are related to metabolic dysregulation and inflammation (e.g., RETN, ITGB2, LEP). Previous investigations [69,70] have noted sex-specific associations between exercise and vascular health, which may, to some degree, be explained by the different associations between markers and activity in the present study. Also, based on SCAPIS data, we have previously shown a borderline interaction between SED/LIPA and sex for coronary atherosclerosis, and the identified markers may be key candidates to explain these differences. Interactions were also noted between activity and waist circumference. For the proteomic markers, all associations were stronger in participants with higher waist circumference (>102 cm for males and >88 cm for females). Limited data is available on the interaction between obesity and PA for the development of cardiometabolic disease, but the present data indicates that associations may be stronger in obese than in normal-weight individuals.

### Strengths and limitations

Strengths of our study include the objective measurement of PA and the examination of lipids and cardiovascular plasma protein biomarkers in association with various characteristics across a range of PA intensities. Moreover, the study includes the discovery and replication of associations between PA and metabolic and cardiovascular protein biomarkers in two separate large, well-characterized population-based cohorts. Further, the study is based on a substantial number of biomarkers analyzed simultaneously using two sensitive, highly specific methods without cross-reactivity and with good agreement compared to other assays. Several limitations balance the strengths of the study. Due to the cross-sectional design, it is not possible to determine the causal direction of the associations. Having or being at high risk for cardiovascular disease could influence the amount of PA. [71] Although accelerometers provide a more robust assessment of PA compared to self-report measures, they are not without limitations. They rely on categorizing the strength of movement (acceleration) rather than directly distinguishing between postures, domains or types of PA. Further, intensities used in this study, and most other studies based on accelerometers, are absolute in the sense that they are the same for individuals with differing work tolerance, or fitness level. As most physiological responses to PA are at least partly related to relative intensity (as a proportion of the individual fitness level), this may constitute a source of misclassification, potentially leading to regression dilution and unidentified associations dependent on fitness levels. Another limitation is the lack of information on ongoing therapies, such as statins or lipid uptake inhibitors. Many of the included variables in the models are subjected to potential residual confounding, i.e.,that a confounding variable is not fully adjusted for, due to imperfect measurement or unwarranted categorization. We have not been able to estimate the potential impact of residual confounding. As in all larger data collections, not all invited subjects choose to participate, leading to potential selection biases. This can

be of major importance when assessing prevalence or estimating reference values. While it still may constitute a potential limitation, it is most probably to a lesser degree when assessing relations within a population such as in these analyses.

The SCAPIS cohort is based on middle-aged Swedish men and women. Although this population is heterogenous in several aspects, the generalizability to other populations is unknown.

## Conclusions

In conclusion, our data suggest differences in the associations between physical activity of various intensity and metabolomic or proteomic biomarkers, in a mixed middle-aged population. These results confirm and expand on previous knowledge, by showing that MVPA and lower-intensity activity have (partly) different relations to CVD risk.

## Supporting information

**S1 Table. Lists of non-significant biomarkers.**
(DOCX)

**S2 Table. Interactions between SED and sex, using model 2 plus an interaction term.**
(DOCX)

**S3 Table. Interactions between MVPA and sex, using model 2 plus an interaction term.**
(DOCX)

**S4 Table. Interactions between SED and waist circumference (dichotomized at 102 cm and 88 cm for male and females, respectively) using model 2 plus an interaction term.**
(DOCX)

**S5 Table. Interactions between MVPA and waist circumference (dichotomized at 102 cm and 88 cm for male and females, respectively) using model 2 plus an interaction term.**
(DOCX)

**S1 Fig. Wenn-diagram showing number of metabolomic biomarkers related to SED, proISED, LIPA and MVPA.** SED encapsuled all biomarkers related to LIPA.
(TIF)

**S2 Fig. Wenn-diagram showing number of proteomic biomarkers related to SED, proISED, LIPA and MVPA.** SED encapsuled all biomarkers related to LIPA.
(TIF)

**S3 Fig. Heat map of relations between metabolomic markers and the PA classes.**
(TIF)

**S4 Fig. Heat map of relations between proteomic markers and the PA classes.**
(TIF)

**S1 Checklist. STROBE-checklist-v4_B4_241209.**
(DOCX)

## Acknowledgments

We would like to acknowledge the help of Biobank Sweden and the local biobank facilities for their services in handling of biological samples and biobanking.

## Author contributions

**Conceptualization:** Örjan Ekblom, Harry Björkbacka, Mats Börjesson, Elin Ekblom-Bak, Carl Johan Östgren.

**Formal analysis:** Örjan Ekblom.

**Investigation:** Örjan Ekblom, Carl Johan Östgren.

**Methodology:** Örjan Ekblom, Harry Björkbacka, Mats Börjesson, Elin Ekblom-Bak, Anders Blomberg, Kenneth Caidahl, Ewa Ehrenborg, Gunnar Engström, Jan Engvall, David Erlinge, Tove Fall, Bruna Gigante, Anders Gummesson, Tomas Jernberg, Lars Lind, David Molnar, Jonas Oldgren, Aidin Rawshani, Johan Sundström, Stefan Söderberg, Patrik Wennberg, Carl Johan Östgren.

**Writing – original draft:** Örjan Ekblom, Harry Björkbacka, Mats Börjesson, Elin Ekblom-Bak, Carl Johan Östgren.

**Writing – review & editing:** Örjan Ekblom, Harry Björkbacka, Mats Börjesson, Elin Ekblom-Bak, Anders Blomberg, Kenneth Caidahl, Ewa Ehrenborg, Gunnar Engström, Jan Engvall, David Erlinge, Tove Fall, Bruna Gigante, Anders Gummesson, Tomas Jernberg, Lars Lind, David Molnar, Jonas Oldgren, Aidin Rawshani, Johan Sundström, Stefan Söderberg, Patrik Wennberg, Carl Johan Östgren.

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
