## [Decision Letter · Decision Letter 0]

28 Jan 2025

Dear Dr. Ekblom,

Thank you for submitting your manuscript to PLOS ONE. After careful consideration, we feel that it has merit but does not fully meet PLOS ONE’s publication criteria as it currently stands. Therefore, we invite you to submit a revised version of the manuscript that addresses the points raised during the review process.

**ACADEMIC EDITOR: **

We look forward to receiving your revised manuscript.

Kind regards,

Hidetaka Hamasaki

Academic Editor

PLOS ONE

2. Thank you for stating the following financial disclosure:  [ÖE was funded by Livförsäkringsbolaget Skandia, Risk&Hälsa.]. At this time, please address the following queries:

Additional Editor Comments (if provided):

Reviewers' comments:

Reviewer's Responses to Questions

**Comments to the Author**

1. Is the manuscript technically sound, and do the data support the conclusions?

Reviewer #1: Yes

Reviewer #2: Yes

Reviewer #3: No

Reviewer #4: Yes

2. Has the statistical analysis been performed appropriately and rigorously?

Reviewer #1: Yes

Reviewer #2: Yes

Reviewer #3: N/A

Reviewer #4: No

3. Have the authors made all data underlying the findings in their manuscript fully available?

Reviewer #1: Yes

Reviewer #2: Yes

Reviewer #3: No

Reviewer #4: No

4. Is the manuscript presented in an intelligible fashion and written in standard English?

Reviewer #1: Yes

Reviewer #2: Yes

Reviewer #3: Yes

Reviewer #4: Yes

Reviewer #1: I would like to congratulate the authors of developing this manuscript which is challenging due to the number of variables and sample size. It is an important information due to unclear association status between accelerometer-based measures of different physical activities and plasma metabolomic and proteomic biomarkers in a population-based study. Many relationship were anticipated, like the association of sedentary with HDL and higher activity level with VLDL and LDL.

It was not clear from the manuscript whether diet plans were selected or advised to the participants and whether their diet could have any differences on the outcomes.

It is worth to be mentioned by authors in the manuscript is the lack of detailed individuals health, medical conditions and therapies which might also affect outcomes like dyslipidemia and being on statin therapy as examples.

There was two short titles for the manuscript, to review this.

Reviewer #2: The manuscript explores an important and timely topic, providing valuable insights into the associations between physical activity and cardiovascular biomarkers. While the study employs decent methodologies and presents significant findings, there are critical areas requiring attention. Addressing these issues will enhance the manuscript's impact. Below, I outline specific critiques for each section of the manuscript.

Title

The title could be more concise. Consider removing redundant phrases like "assessed physical activity variables" to streamline it without losing clarity (e.g., "Associations between Physical Activity and CVD-related Biomarkers").

Abstract

Line 71–98: The terminology used (e.g., "mirroring," "robust relations") might be too technical for a general audience. Brief definitions or rephrasing could help.

The inclusion of exact numerical values for validated biomarkers might overwhelm the abstract's narrative. Consider summarizing these findings qualitatively.

The abstract fails to explicitly state the study's limitations, which would provide a more balanced overview.

Introduction

Lines 100–133: There is an overreliance on citing older studies without critical analysis of gaps they leave. For example, it would be helpful to contrast self-reported physical activity data with accelerometer-based data more thoroughly.

The aim of the study (line 130–133) lacks a hypothesis-driven narrative. Stating a clear research question would enhance focus.

Methods

Line 136–243:

The description of accelerometer processing could be expanded to address potential biases introduced by the low-frequency extension filter.

No mention is made of how outliers or missing data were handled statistically (e.g., imputation methods).

Confounding variables (lines 203–214): While a comprehensive list is provided, there is no discussion on whether any were prioritized based on their theoretical significance.

Results

Line 254–305:

Statistical findings are often presented without adequate contextualization. For instance, what are the implications of HDL species being negatively associated with SED? A brief interpretation in the results section could aid clarity.

The authors mention "robust relations" but do not adequately define how this robustness was assessed beyond statistical significance.

Table 1: Some demographic variables (e.g., education) are reported without discussing their impact on the study findings.

Discussion

Line 307–319:

The discussion is overly descriptive and does not sufficiently explore the causal implications of the findings. For instance, are the observed associations indicative of mechanistic pathways or simply correlations?

Potential limitations like the cross-sectional nature of the study are mentioned but downplayed. A more critical acknowledgment is warranted, particularly around confounding and selection biases.

No mention is made of generalizability. Since participants are middle-aged Swedes, findings may not apply to diverse populations.

Reviewer #3: The focus of this study by Ekblom et. al. is to find an association between various intensity PA classes and metabolomic and cardiovascular protein biomarkers in a middle-aged population.

The main drawback of this study is that it is not giving any new information except expanding on the previous correlation between these variables. Metabolomic and Proteomic Profiling are done in different conditions and cohort which doesn't give much emphasis to the manuscript.

Reviewer #4: I am grateful for the opportunity to review this manuscript, which examined the associations between accelerometer-assessed physical activity behaviours and CVD-related metabolomic and proteomic biomarkers using two cross-sectional data from the main SCAPIS cohort (discovery data) and SCAPIS pilot cohort (validation data). Though the manuscript is well-written and contains some interesting findings, a few issues may have impacted the scientific quality of the findings.

Below are some suggestions/comments/questions the authors may consider if they find them intuitive to enhance the quality of the paper.

Abstract:

1. The authors could briefly describe the demographic characteristics of the studied participants, both the discovery and validated samples.

2. The number of the main SCAPIS cohort (discovery = 5557) does not match the description in the main text – discovery = 4647)

3. The authors can consider mentioning the analytical method used to examine the associations in the abstract.

4. The description of the observed associations is unclear, and the authors can help the readers by clarifying the metabolic and proteomic biomarkers and the direction of their associations with the physical activity behaviour classes, especially for the proteomic markers (lines 90 – 92).

Introduction:

1. The authors made several factual statements in the first four opening sentences (page 5, lines 100 – 104) and it would be more appropriate for them to provide some supporting citations (references).

2. Similarly, the sentences on page 5 lines 110 – 114 may need some supporting references.

3. Though the introduction looks well-written, it lacks some relevant and specific details that justify the authors’ decision to examine the relationships of physical activity behaviour classes with metabolic and proteomic biomarkers. There is extensive evidence of the associations between the physical activity behaviour spectrum and biomarkers. A brief contextual review of the evidence on moderate-to-vigorous intensity physical activity (MVPA), light-intensity physical activity (LIPA) and sedentary behaviour (SB) (including SB bouts), and the interdependence of these behaviours, would have been appropriate to build their study rationale.

Methods:

1. The description on page 9, lines 181 – 184 is a little confusing. The authors may consider clarifying the total number of discovery samples from the regions/sites (5075) and the initial number of included subjects (30154) and after excluding those with missing data the total number of remaining subjects (4887).

2. It is not clear what “FA variables” means. It may presumably be a typo (PA – physical activity); if it isn’t, it would be appropriate to define abbreviations when used for the first time in the document.

3. It would be appropriate to define the abbreviation “E-GFR”.

4. Some readers may also be interested in how the biomarkers used in the Cockcroft-Gault Equation to measure the estimated glomerular filtration rate (eGFR) were assessed.

Results:

1. The analysis and results are complex, and the descriptions are comprehensive. There are a few instances, though, where the authors would have been more specific about the direction of the associations (beneficial ‘+’ or detrimental ‘-’) of the physical activity behaviours with the biomarkers. For instance, the description of the associations on page 27, line 298.

2. The sentence “Opposite relations were found for ITGB2 and LPL” on page 27, line 316 needs clarity.

3. It is impressive that the authors also explored the potential non-linear associations; however, describing the non-linear analytic approach utilised in the statistical analysis section would be more appropriate.

4. Did the authors also examine the non-linear associations with the metabolic/proteomic biomarkers of LIPA and prolonged sedentary behaviour?

5. There seem to be gender differences in the associations, what is not clear, though, is the direction of the associations. The authors may consider describing the direction of the physical activity behaviours and sex interaction associations with the biomarkers.

6. Similarly, the authors could describe the direction of the physical activity behaviours and waist circumference interaction associations with the biomarkers.

7. Did the authors also examine the interaction of LIPA with sex or waist circumference?

8. Association tables: the authors may have a specific reason to only report coefficients. However, reporting coefficients with confidence intervals or standard errors would be more informative in understanding the statistical significance of the findings.

Discussion:

1. The findings described in the results mainly focus on the directions rather than the magnitudes/strengths of the associations, however, the authors seem to be highlighting the strengths of the associations in their discussion of the findings.

2. There are several factual statements in the discussion that may need appropriate supporting evidence (references). For example, the sentence on page 34, lines 454 – 457.

Figures:

1. The authors did not adequately justify why LIPA was excluded from the diagram. As the authors noted in the results section LIPA and sedentary behaviour have a strong negative interrelation, some readers may find it informative if all the physical activity behaviours, including LIPA, are shown in the vain diagram with the intersections.

2. The authors could consider a brief description of the vain diagram, highlighting the intersections in the legend.

**Do you want your identity to be public for this peer review?** For information about this choice, including consent withdrawal, please see our Privacy Policy

Reviewer #1: No

Reviewer #2: No

Reviewer #3: No

Reviewer #4: No

---

## [Author Response · Author response to Decision Letter 1]

12 Mar 2025

Journal requirements

Comment Author response Alterations

1. Please ensure that your manuscript meets PLOS ONE's style requirements, including those for file naming. The PLOS ONE style templates can be found at We have compared requirements including naming.

2. Thank you for stating the following financial disclosure: [ÖE was funded by Livförsäkringsbolaget Skandia, Risk&Hälsa.]. At this time, please address the following queries:

a) Please clarify the sources of funding (financial or material support) for your study. List the grants or organizations that supported your study, including funding received from your institution. Thanks for pointing this out. This information has been added to the submission system. In the manuscript, this information has been moved from Acknowledgements to Funding. Text updated on p 39, under "Acknowledgements" and "Funding"

b) State what role the funders took in the study. If the funders had no role in your study, please state: “The funders had no role in study design, data collection and analysis, decision to publish, or preparation of the manuscript.” As the funding bodies did not have any such role, the statement has been added to the submission system and the manuscript. During the work on this paper, the SCAPIS office launched an altered standard wording for Acknowledgements, which has now been added Text updated on p 39, under "Acknowledgements" and "Funding"

c) If any authors received a salary from any of your funders, please state which authors and which funders. The main author was funded by a grant from Livförsäkringsbolaget Skandia in the form of salary. This has been added to the manuscript Added text on p 39 under "Funding"

We note that you have indicated that there are restrictions to data sharing for this study. For studies involving human research participant data or other sensitive data, we encourage authors to share de-identified or anonymized data. However, when data cannot be publicly shared for ethical reasons, we allow authors to make their data sets available upon request. For information on unacceptable data access restrictions, please see http://journals.plos.org/plosone/s/data-availability#loc-unacceptable-data-access-restrictions.

Data used in this paper are a part of the SCAPIS database. Researcher aiming to obtain access to these data may apply for them to the data access committee at the SCAPIS office. The data access agreement between the authors and the SCAPIS hinders any direct sharing of data. Further, the ethical permissions for this paper do not allow for direct sharing of de-identified data. The data access committee can be reached at scapis@scapis.org, with more information at the website www.scapis.org.

b) If there are no restrictions, please upload the minimal anonymized data set necessary to replicate your study findings to a stable, public repository and provide us with the relevant URLs, DOIs, or accession numbers. Please see http://www.bmj.com/content/340/bmj.c181.long for guidelines on how to de-identify and prepare clinical data for publication. For a list of recommended repositories, please see https://journals.plos.org/plosone/s/recommended-repositories. You also have the option of uploading the data as Supporting Information files, but we would recommend depositing data directly to a data repository if possible. n/a

Please update your Data Availability statement in the submission form accordingly. Data Availability statement updated aThe following text has been added: The General Data Protection Regulation (EU 2016/679) classifies de-identified versions of sensitive data that are sufficiently detailed to allow for re-identification as sensitive personal information. According to Swedish law (Law 2003:460 for ethical review of research involving humans), ethical permission is required to process such data. In accordance with Swedish legislation, the data can and will be made available to researchers who meet the criteria for access to confidential data, which includes obtaining their own ethics approval from the Swedish Ethical Review Authority (email: registrator@etikprovning.se; website: https://etikprovningsmyndigheten.se). Data applications can then be made by contacting SCAPIS (email: scapis@scapis.org; website: https://www.scapis.org/data-access/).

Reviewer #1

I would like to congratulate the authors of developing this manuscript which is challenging due to the number of variables and sample size. It is an important information due to unclear association status between accelerometer-based measures of different physical activities and plasma metabolomic and proteomic biomarkers in a population-based study. Many relationship were anticipated, like the association of sedentary with HDL and higher activity level with VLDL and LDL. We agree that the number of biomarkers in this paper is large, but also constitute a strength of the paper and that some of the relationships were expected. None.

It was not clear from the manuscript whether diet plans were selected or advised to the participants and whether their diet could have any differences on the outcomes. No diet plan was given, and diet was not controlled other than that subjects were in a fasted state during blood sampling. Food frequency questionnaires were used to survey the dietary habits of the participants. Data on macronutrient intake were included in the analyses. Altered text on p 11, rows, 247-248

It is worth to be mentioned by authors in the manuscript is the lack of detailed individuals health, medical conditions and therapies which might also affect outcomes like dyslipidemia and being on statin therapy as examples. We agree that there are conditions that may be of relevance that we could not include, such as those mentioned. However, we chose to include overweight status, kidney function, diabetes status, along with behaviours such as diet and smoking. We have included a paragraph on this in the limitations section, on p 37, rows 546-548

There was two short titles for the manuscript, to review this. We apologize, this has been adjusted. The latter short title has been removed.

Reviewer #2

The manuscript explores an important and timely topic, providing valuable insights into the associations between physical activity and cardiovascular biomarkers. While the study employs decent methodologies and presents significant findings, there are critical areas requiring attention. Addressing these issues will enhance the manuscript's impact. Below, I outline specific critiques for each section of the manuscript. We thank the reviewer for the time taken to review our manuscript and the valuable input.

The title could be more concise. Consider removing redundant phrases like "assessed physical activity variables" to streamline it without losing clarity (e.g., "Associations between Physical Activity and CVD-related Biomarkers"). Thank you for the input. We agree that the title can be perceived as bulky. Title shortened for clarity.

Line 71–98: The terminology used (e.g., "mirroring," "robust relations") might be too technical for a general audience. Brief definitions or rephrasing could help. We agree and have tried to clarify this further.

Alterations in results section of abstract.

The inclusion of exact numerical values for validated biomarkers might overwhelm the abstract's narrative. Consider summarizing these findings qualitatively. We agree with the reviewer that this could be less detailed. However, we have discussed this and tried to express the results in overarching terms, without being too vague. We hope that this way of expressing the results might work in many readers minds. none

Lines 100–133: There is an overreliance on citing older studies without critical analysis of gaps they leave. For example, it would be helpful to contrast self-reported physical activity data with accelerometer-based data more thoroughly. We agree with the reviewer that this is most important. However, to our knowledge, there are no recent analysis that compare the predictive validity on biomarkers between self-reports and motion sensors. We therefore chose to base a discussion on this on an earlier paper and a few newer on similar topics. Addition of discussion and references on rows 127-130.

The aim of the study (line 130–133) lacks a hypothesis-driven narrative. Stating a clear research question would enhance focus. We agree with the reviewer that our aim is not hypothesis driven, but rather explorative. The large number of biomarkers studied makes it difficult to synthesise a single hypothesis, but based on studies presented in the background, it would be plausible to state that we believe that the relations would be different across PA classes. On rows 165-166, we state We hypothesize that relations between PA and biomarkers are different across PA classes.

The description of accelerometer processing could be expanded to address potential biases introduced by the low-frequency extension filter. The low-frequency extension (LFE) filter may to some extent affect time spent in SED or in prolSED, but not the other PA classes. The relations to the studied biomarkers are assumed to be more stable. We have added a short discussion on this on rows 200-201.

No mention is made of how outliers or missing data were handled statistically (e.g., imputation methods).

We agree that this could be described in more detail. No imputation was applied as stated on row 235, but the text is now slightly more detailed. The extent of missing data is given under “Blood sampling and analysis”. We state that “No imputation was applied for missing data.” on row 265-266.

Confounding variables (lines 203–214): While a comprehensive list is provided, there is no discussion on whether any were prioritized based on their theoretical significance. We acknowledge this shortcoming to the text. The included independent variables all have their theoretical significance, and it is difficult to rank them due to the varying nature of the different outcome variables. However, we considered it more transparent to apply two models. Age, sex, kidney function and diabetes function (and plate number) were thought as variables suitable to adjust for in the first model. In model 2, remaining variables were included, due to their potential influence on the relation between PA and biomarker level.

We have added a short part on model development.

Added sentence on row 282-283

Statistical findings are often presented without adequate contextualization. For instance, what are the implications of HDL species being negatively associated with SED? A brief interpretation in the results section could aid clarity. We agree that there are many relations worth discussing. The implications are many and we have tried to elaborate on this in the discussion. However, the cross-sectional design limits the ability to discuss implications. Added short sentence on row 453-454

The authors mention "robust relations" but do not adequately define how this robustness was assessed beyond statistical significance. Under “Statistics” we define “robust” relations as those with opposing relations to MVPA vs SED/prolSED and to separate those from biomarkers with relations to only one PA-class. The idea is to separate biomarkers with a more solid (“robust”) and more discrete (“separate”) relation to PA beyond the application of two-sample validation and two different regression models.

We agree that this could be further described, and we have altered the description in the text for clarity. Added sentence on rows 288-290

Table 1: Some demographic variables (e.g., education) are reported without discussing their impact on the study findings. We thank the reviewer for this comment. The results from the regression analyses including the other independent variables than PA could be presented and discussed. However, to keep the focus on the aim of the paper, we chose to report only the results for the PA classes.

The discussion is overly descriptive and does not sufficiently explore the causal implications of the findings. For instance, are the observed associations indicative of mechanistic pathways or simply correlations? We agree that the discussion refrains from discussing in causal terms, which is due to the study design. However, the reviewer is correct in that causality could be discussed to a certain degree, based on potential physiological mechanisms. For metabolomic markers, PA is known to increase lipid metabolism and thereby could affect lipid profile via LPL activation or (up-)regulation of LDL receptors. Proteomic markers related to inflammation (such as IL-6) may be affected by PA by alterations to the autonomic system, AMPK activation or changes in for example visceral fat deposits. We have added a few examples, without arguing that we have covered all such mechanisms.

We deemed the number of such mechanisms to be too high to be systematically covered in the discussion. Added examples of potential physiological pathways on rows 504-510.

Potential limitations like the cross-sectional nature of the study are mentioned but downplayed. A more critical acknowledgment is warranted, particularly around confounding and selection biases. We agree with the reviewer that the cross-sectional design prevents any causal inferences, and we have mentioned this throughout the manuscript. We have added a section on selection biases and possible residual confounding in the “Strength and limitations”-section. Added a paragraph on this on rows 546-555

No mention is made of generalizability. Since participants are middle-aged Swedes, findings may not apply to diverse populations. We agree with the reviewer and have added a text on this Added text on rows 556-558

Reviewer #3

The focus of this study by Ekblom et. al. is to find an association between various intensity PA classes and metabolomic and cardiovascular protein biomarkers in a middle-aged population.

The main drawback of this study is that it is not giving any new information except expanding on the previous correlation between these variables. Metabolomic and Proteomic Profiling are done in different conditions and cohort which doesn't give much emphasis to the manuscript.

We agree with the reviewer that previous investigations have described several of these relations. However, we believe that the study including sensor-based PA assessments and cross-validation in a relatively large and heterogenous population has a value in describing the wider patterns of relations. Also, the lack of convincing relationships between prolSED and biomarkers outside of those found for SED is of potential importance for the design of future intervention studies.

Reviewer #4

I am grateful for the opportunity to review this manuscript, which examined the associations between accelerometer-assessed physical activity behaviours and CVD-related metabolomic and proteomic biomarkers using two cross-sectional data from the main SCAPIS cohort (discovery data) and SCAPIS pilot cohort (validation data). Though the manuscript is well-written and contains some interesting findings, a few issues may have impacted the scientific quality of the findings.

Below are some suggestions/comments/questions the authors may consider if they find them intuitive to enhance the quality of the paper.

We thank the reviewer for critically assessing our manuscript and the possibility to revise in accordance

---

## [Decision Letter · Decision Letter 1]

28 Mar 2025

Dear Dr. Ekblom,

Thank you for submitting your manuscript to PLOS ONE. After careful consideration, we feel that it has merit but does not fully meet PLOS ONE’s publication criteria as it currently stands. Therefore, we invite you to submit a revised version of the manuscript that addresses the points raised during the review process.

We look forward to receiving your revised manuscript.

Kind regards,

Hidetaka Hamasaki

Academic Editor

PLOS ONE

Reviewers' comments:

Reviewer's Responses to Questions

**Comments to the Author**

Reviewer #1: All comments have been addressed

Reviewer #2: All comments have been addressed

Reviewer #4: All comments have been addressed

2. Is the manuscript technically sound, and do the data support the conclusions?

Reviewer #1: Partly

Reviewer #2: Yes

Reviewer #4: Partly

3. Has the statistical analysis been performed appropriately and rigorously?

Reviewer #1: Yes

Reviewer #2: Yes

Reviewer #4: Yes

4. Have the authors made all data underlying the findings in their manuscript fully available?

Reviewer #1: Yes

Reviewer #2: Yes

Reviewer #4: No

5. Is the manuscript presented in an intelligible fashion and written in standard English?

Reviewer #1: Yes

Reviewer #2: Yes

Reviewer #4: Yes

Reviewer #1: The authors have adequately addressed my comments raised in the previous review and I don't have any additional comments.

Reviewer #2: I would like to congratulate and thank the author for carefully addressing my comments. I have no further concerns.

Reviewer #4: The authors did a great job satisfactorily addressing most of the concerns; however, their response to one of the comments raises a major concern about the credibility of the reported findings. The reported effect sizes look rather very small. The tables do not look that huge, and the authors may consider restructuring the tables. For them to only report the significant findings without confidence intervals or standard errors may indicate that the authors were selective in their reporting and compromised the validity of the findings. The question, though, is why the authors did not report non-significant results. In some points, the authors also highlight the strength of the associations in their discussion, which is not obvious in the reported coefficients without confidence intervals. The comment with the response is below:

Comment: Association tables: the authors may have a specific reason to only report coefficients. However, reporting coefficients with confidence intervals or standard errors would be more informative in understanding the statistical significance of the findings.

Response: We agree that adding CIs or measures of variance would give more information. However, the tables are rather large as is and contain only the significant relations. Confidence intervals would all exclude “zero”. The magnitude of for example standard deviations would be small and difficult to read. Therefore, we chose not to include these data.

**Do you want your identity to be public for this peer review?** For information about this choice, including consent withdrawal, please see our Privacy Policy

Reviewer #1: No

Reviewer #2: No

Reviewer #4: No

---

## [Author Response · Author response to Decision Letter 2]

30 Apr 2025

Reviewer #4:

1. The authors did a great job satisfactorily addressing most of the concerns; however, their response to one of the comments raises a major concern about the credibility of the reported findings. The reported effect sizes look rather very small. The tables do not look that huge, and the authors may consider restructuring the tables. For them to only report the significant findings without confidence intervals or standard errors may indicate that the authors were selective in their reporting and compromised the validity of the findings.

2. The question, though, is why the authors did not report non-significant results.

3. In some points, the authors also highlight the strength of the associations in their discussion, which is not obvious in the reported coefficients without confidence intervals. The comment with the response is below:

[Citing earlier conversations]

Comment: Association tables: the authors may have a specific reason to only report coefficients. However, reporting coefficients with confidence intervals or standard errors would be more informative in understanding the statistical significance of the findings.

Response: We agree that adding CIs or measures of variance would give more information. However, the tables are rather large as is and contain only the significant relations. Confidence intervals would all exclude “zero”. The magnitude of for example standard deviations would be small and difficult to read. Therefore, we chose not to include these data.

Author response:

1. We agree with the reviewer that the coefficients are nominally low, which was mentioned in the manuscript. Based on the suggestion from the reviewer, we have added SE to tables 2, 3, and 4, which may help in interpreting the values.

2. We agree with the reviewer that it is essential to report null findings in scientific reports and are therefore listing biomarkers with non-significant relations to the different PA classes in supplementary table 5.

3. The addition of SEs to the tables may help interpretation in this instance. However, we regarded all significant relations as essential and chose to mention the significant biomarkers with the nominally largest coefficient instead of, for example, those with the lowest p-value. Thus note, all markers in tables 2-4 are significant.

---

## [Decision Letter · Decision Letter 2]

18 May 2025

Associations between physical activity and CVD-related metabolomic and proteomic biomarkers.

PONE-D-24-56786R2

Dear Dr. Ekblom,

We’re pleased to inform you that your manuscript has been judged scientifically suitable for publication and will be formally accepted for publication once it meets all outstanding technical requirements.

Kind regards,

Hidetaka Hamasaki

Academic Editor

PLOS ONE

Additional Editor Comments (optional):

Thank you for resubmitting the revised manuscript.

On behalf of Reviewer 4, I have reviewed your responses and the corresponding revisions. Regarding the comment about the small effect size, you have appropriately provided the standard errors. As for the biomarkers that did not show a significant association, they are now presented in Supplementary Table 5. These revisions sufficiently address the concerns raised by Reviewer 4. Therefore, I am pleased to inform you that your manuscript is accepted for publication. Congratulations, and thank you for reporting such a valuable piece of research.
---

## [Editor Report · Acceptance letter]

PONE-D-24-56786R2

PLOS ONE

Dear Dr. Ekblom,

I'm pleased to inform you that your manuscript has been deemed suitable for publication in PLOS ONE. Congratulations! Your manuscript is now being handed over to our production team.

Kind regards,

on behalf of

Dr. Hidetaka Hamasaki

Academic Editor

PLOS ONE